# Neural Anisotropy Directions

**Guillermo Ortiz-Jiménez**[*]
EPFL, Lausanne, Switzerland
guillermo.ortizjimenez@epfl.ch

**Apostolos Modas**[*]
EPFL, Lausanne, Switzerland
apostolos.modas@epfl.ch

**Seyed-Mohsen Moosavi-Dezfooli**
ETH Zürich, Zurich, Switzerland
seyed.moosavi@inf.ethz.ch

**Pascal Frossard**
EPFL, Lausanne, Switzerland
pascal.frossard@epfl.ch

## Abstract

In this work, we analyze the role of the network architecture in shaping the inductive bias of deep classifiers. To that end, we start by focusing on a very simple problem, i.e., classifying a class of linearly separable distributions, and show that, depending on the direction of the discriminative feature of the distribution, many state-of-the-art deep convolutional neural networks (CNNs) have a surprisingly hard time solving this simple task. We then define as *neural anisotropy directions (NADs)* the vectors that encapsulate the directional inductive bias of an architecture. These vectors, which are specific for each architecture and hence act as a signature, encode the preference of a network to separate the input data based on some particular features. We provide an efficient method to identify NADs for several CNN architectures and thus reveal their directional inductive biases. Furthermore, we show that, for the CIFAR-10 dataset, NADs characterize the features used by CNNs to discriminate between different classes.

## 1 Introduction

In machine learning, given a finite set of samples, there are usually multiple solutions that can perfectly fit the training data, but the *inductive bias* of a learning algorithm selects and prioritizes those solutions that agree with its *a priori* assumptions [1, 2]. Arguably, the main success of deep learning has come from embedding the right inductive biases in the architectures, which allow them to excel at tasks such as classifying ImageNet [3], understanding natural language [4], or playing Atari [5].

Nevertheless, most of these biases have been generally introduced based on heuristics that rely on generalization performance to *naturally select* certain architectural components. As a result, although these deep networks work well in practice, we still lack a proper characterization and a full understanding of their actual inductive biases. In order to extend the application of deep learning to new domains, it is crucial to develop generic methodologies to systematically identify and manipulate the inductive bias of deep architectures.

Towards designing architectures with desired properties, we need to better understand the bias of the current networks. However, due to the co-existence of multiple types of inductive biases within a neural network, such as the preference for simple functions [6], or the invariance to certain group transformations [7], identifying all biases at once can be challenging. In this work, we take a bottom-up stance and focus on a fundamental bias that arises in deep architectures even for classifying linearly

---

[*]Equal contribution. Correspondence to {guillermo.ortizjimenez, apostolos.modas}@epfl.ch. The code to reproduce our experiments can be found at https://github.com/LTS4/neural-anisotropy-directions.

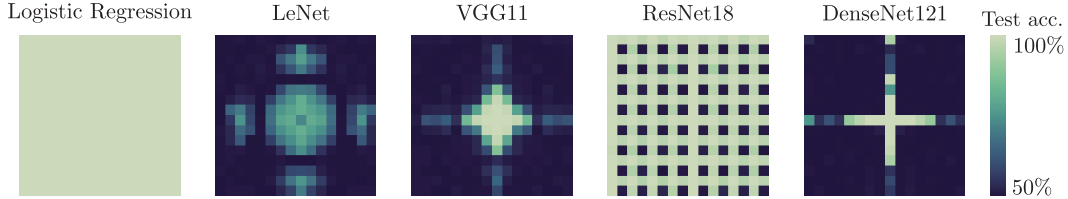

Figure 1: Test accuracies of different architectures [8–11]. Each pixel corresponds to a linearly separable dataset (with $10,000$ training samples) with a single discriminative feature aligned with a basis element of the 2D-DFT. We use the standard 2D-DFT convention and place the dataset with lower discriminative frequencies at the center of the image, and the higher ones extending radially to the corners. All networks (except LeNet) achieve nearly $100\%$ train accuracy. ($\sigma = 3, \epsilon = 1$)

separable datasets. In particular, we show that depending on the nature of the dataset, some deep neural networks can only perform well when the discriminative information of the data is aligned with certain directions of the input space. We call this bias the *directional inductive bias* of an architecture.

This is illustrated in Fig. 1 for state-of-the-art CNNs classifying a set of linearly separable distributions with a single discriminative feature lying in the direction of some Fourier basis vector[1]. Remarkably, even the gigantic DenseNet [8] only generalizes to a few of these distributions, despite common belief that, due to their superior capacity, such networks can learn most functions efficiently. Yet, even a simple logistic regression eclipses their performance on a simple linearly separable task.

In this paper, we aim to explain why this happens, and try to understand why some linear distributions are easier to classify than others. To that end, we introduce the concept of *neural anisotropy directions* to characterize the directional inductive bias of an architecture.

**Definition 1** (Neural anisotropy directions). *The neural anisotropy directions (NADs) of a specific architecture are the ordered set of orthonormal vectors $\{u_i\}_{i=1}^{D}$ ranked in terms of the preference of the network to separate the data in those particular directions of the input space.*

In general, though, quantifying the preference of a complex network to separate data in certain directions is not straightforward. In this paper, we will show that measuring the performance of a network on different versions of a linearly separable dataset can reveal its directional inductive bias. Yet, we will provide an efficient computational method to fully characterize this bias in terms of NADs, independent of training data. Finally, we will reveal that NADs allow a network to prioritize certain discriminating features of a dataset, and hence act as important conductors of generalization.

Our main contributions can be summarized as follows:

- We characterize the directional inductive bias in the spectral domain of state-of-the-art CNNs, and explain how pooling layers are a major source for this bias.

- More generally, we introduce a new efficient method to identify the NADs of a given architecture using only information available at initialization.

- Finally, we show that the importance of NADs is not limited to linearly separable tasks, and that they determine the selection of discriminative features of CNNs trained on CIFAR-10.

We believe that our findings can impact future research in novel architectures, by allowing researchers to compare and quantify the specific inductive bias of different networks.

**Related work** The inductive bias of deep learning has been extensively studied in the past. From a theoretical point of view, this has mainly concerned the analysis of the implicit bias of gradient descent [6, 12–15], the stability of convolutional networks to image transformations [7, 16], or the impossibility of learning certain combinatorial functions [17, 18]. Related to our work, it has been shown that some heavily overparameterized neural networks will provably learn a linearly separable distribution when trained using stochastic gradient descent (SGD) [19]. This result, however, only applies to the case of neural networks with two fully connected layers, and it says little about the

learnability of linearly separable distributions with complex architectures. On the practical side, the myriad of works that propose new architectures are typically motivated by some informal intuition on their effect on the inductive bias of the network [8–11, 20–22]. Although little attention is generally given to properly quantifying these intuitions, some works have recently analyzed the role of architecture in the translation equivariance of modern CNNs [23–25].

## 2    Directional inductive bias

We will first show that the test accuracy on different versions of a linearly separable distribution can reveal the directional inductive bias of a network towards specific directions. In this sense, let $\mathcal{D}(\boldsymbol{v})$ be a linearly separable distribution parameterized by a unit vector $\boldsymbol{v} \in \mathbb{S}^{D-1}$, such that any sample $(\boldsymbol{x}, y) \sim \mathcal{D}(\boldsymbol{v})$ satisfies $\boldsymbol{x} = \epsilon y \boldsymbol{v} + \boldsymbol{w}$, with noise $\boldsymbol{w} \sim \mathcal{N}\left(\boldsymbol{0}, \sigma^2(\boldsymbol{I}_D - \boldsymbol{v}\boldsymbol{v}^T)\right)$ orthogonal to the direction $\boldsymbol{v}$, and $y$ sampled from $\{-1, +1\}$ with equal probability. Despite $\mathcal{D}(\boldsymbol{v})$ being linearly separable based on $\boldsymbol{v}$, note that if $\epsilon \ll \sigma$ the noise will dominate the energy of the samples, making it hard for a classifier to identify the generalizing information in a finite-sample dataset.

In practice, it is not feasible to test the performance of a classifier on all possible versions of $\mathcal{D}(\boldsymbol{v})$. Nevertheless, one can at least choose a spanning basis of $\mathbb{R}^D$, from where a set of possible directions $\{\boldsymbol{v}_i\}_{i=1}^D$ can be picked. Informally speaking, if a direction is aligned with the inductive bias of the network under study, then its performance on $\mathcal{D}(\boldsymbol{v})$ would be very good. Otherwise, it would be bad.

We validate our hypothesis on common CNNs used for image classification with a $32 \times 32$ single-channel input. We use the two-dimensional discrete Fourier basis (2D-DFT) – which offers a good representation of the features in standard vision datasets [26–28] – to generate the selected vectors[2]. The difference in performance on these experiments underlines the strong bias of these networks towards certain frequency directions (see Fig. 1). Surprisingly, beyond test accuracy, the bias can also be identified during training, as it takes much longer to converge for some data distributions than others, even when they have little noise (see Fig. 2). This is, the directional inductive bias also plays a role in optimization.

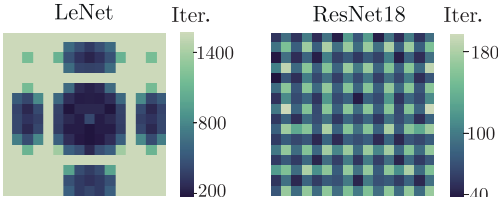

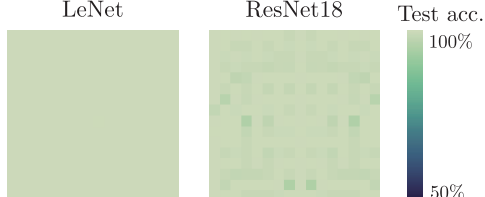

Figure 2: Training iterations required to achieve a small training loss on different $\mathcal{D}(\boldsymbol{v}_i)$ aligned with some Fourier basis vectors ($\sigma = 0.5$).

Figure 3: Test accuracies on different $\mathcal{D}(\boldsymbol{v}_i)$ aligned with some Fourier basis vectors when removing all pooling layers.

One plausible explanation for these results is that under multiple possible fitting solutions on a finite dataset, the network prioritizes those that are aligned with certain frequency directions. Therefore, if the prioritized solutions align with the discriminative features of the data, the classifier can easily generalize. Otherwise, it just overfits to the training data. Finally, we can note the diverse patterns in Fig 1 and Fig 2. CNNs are composed of many modules, and their interconnection can shape the inductive bias of the network in complex ways. In particular, as seen in Fig. 3, if we remove pooling from these networks (with fully connected layers properly adjusted) their performance on different frequencies is equalized. Pooling has previously been shown to modulate the inductive bias of CNNs in the spatial domain [24]; however, it seems that it does so in the spectral domain, as well. This also confirms that the overfitting of these models on this naïve distribution cannot simply be due to their high complexity, as removing pooling technically increases their capacity, and yet their test accuracy improves.

In general, a layer in the architecture can shape the bias in two main ways: by causing an anisotropic loss of information, or by anisotropically conditioning the optimization landscape. In what follows we describe each of them and illustrate their effect through the example of a linear pooling layer.

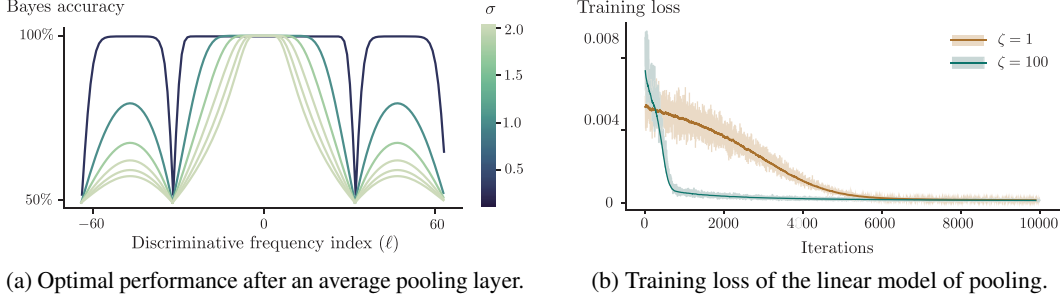

(a) Optimal performance after an average pooling layer.

(b) Training loss of the linear model of pooling.

Figure 4: Effects of pooling ($S = 4, D = 128$) on the inductive bias of a deep neural network.

## 2.1 Anisotropic loss of information

We refer to an anisotropic loss of information as the result of any transformation that harms generalization in specific directions, e.g., by injecting noise with an anisotropic variance. Under these circumstances, any information in the noisy directions will not be visible to the network.

Let $\hat{\boldsymbol{x}} = \mathcal{F}(\boldsymbol{x})$ denote the Fourier transform of an input vector $\boldsymbol{x}$ entering a linear pooling layer (e.g., average pooling) with a subsampling factor $S$. Without loss of generality, let $(\boldsymbol{x}, y) \sim \mathcal{D}(\boldsymbol{v}_\ell)$, and $\boldsymbol{v}_\ell = \mathcal{F}^{-1}(\boldsymbol{e}_\ell)$ with $\boldsymbol{e}_\ell$ representing the $\ell$-th canonical basis vector of $\mathbb{R}^D$. Then, the Fourier transform of the output of the pooling layer satisfies $\hat{\boldsymbol{z}} = \boldsymbol{A}(\hat{\boldsymbol{m}} \odot \hat{\boldsymbol{x}})$ where $\boldsymbol{A} \in \mathbb{R}^{M \times D}$ represents an aliasing matrix such that $\boldsymbol{A} = \frac{1}{\sqrt{S}} \begin{bmatrix} \boldsymbol{I}_M & \cdots & \boldsymbol{I}_M \end{bmatrix}$ with $M = \lceil D/S \rceil$. Here, $\hat{\boldsymbol{m}} \odot \hat{\boldsymbol{x}}$ is the representation in the spectral domain of the convolution of a prefilter $\hat{\boldsymbol{m}}$, e.g., average filtering, with the input signal. Expanding this expression, the spectral coefficients of the output of pooling become

$$\hat{\boldsymbol{z}}[t] = \frac{1}{\sqrt{S}} \sum_{k=0}^{S-1} \hat{\boldsymbol{m}}[\![k \cdot M + t]\!]_D \; \hat{\boldsymbol{x}}[\![k \cdot M + t]\!]_D, \tag{1}$$

where $\hat{\boldsymbol{x}}[\![i]\!]_D$ represents the $(i \bmod D)$-th entry of $\hat{\boldsymbol{x}}$. The following theorem expresses the best achievable performance of any classifier on the distribution of the output of pooling.

**Theorem 1** (Bayes optimal classification accuracy after pooling). *After pooling, the best achievable test classification accuracy on the distribution of samples drawn from $\mathcal{D}(\boldsymbol{v}_\ell)$ can be written as*

$$1 - \mathcal{Q}\left(\frac{\sqrt{2}\epsilon}{2\sigma}\gamma(\ell)\right) \quad \text{with} \quad \gamma^2(\ell) = \frac{|\hat{\boldsymbol{m}}[\ell]|^2 \cdot S}{\sum_{k=1}^{S-1} |\hat{\boldsymbol{m}}[\![\ell + k \cdot M]\!]_D|^2}, \tag{2}$$

*and $\mathcal{Q}(\cdot)$ representing the tail distribution function of the standard normal distribution.*

*Proof.* See Sec. B.1 of Supp. material. □

The intuition behind this theorem lies in (1). Note that after pooling the discriminative information appears only at position $[\![\ell]\!]_M$ and that its signal-to-noise ratio $\gamma(\ell)$ is completely characterized by $\ell$. For this reason, we say that pooling acts as an anisotropic lossy information channel (see Fig 4a).

## 2.2 Anisotropic conditioning of the optimization landscape

Even if there is no information loss, the dependency of the optimization landscape on the discriminative direction can cause a network to show some bias towards the solutions that are better conditioned. This phenomenon can happen even on simple architectures. Hence, we illustrate it by studying an idealized deep linear model of the behaviour of pooling in the spectral domain.

In particular, we study the network $f_{\boldsymbol{\theta}, \boldsymbol{\phi}}(\boldsymbol{x}) = \boldsymbol{\theta}^T \boldsymbol{A}(\boldsymbol{m} \odot \boldsymbol{\phi} \odot \boldsymbol{x})$. In this model, $\boldsymbol{\phi}$ plays the role of the spectral response of a long stack of convolutional layers, $\boldsymbol{m}$ the spectral response of the pooling prefilter, and $\boldsymbol{\theta}$ the parameters of a fully connected layer at the output.

For the sake of simplicity, we assume that the data follows $\boldsymbol{x} = \epsilon y \boldsymbol{e}_\ell + \boldsymbol{w}$ with isotropic noise $\boldsymbol{w} \sim \mathcal{N}(\boldsymbol{0}, \sigma^2 \boldsymbol{I}_D)$. Note that for this family of datasets, the best achievable performance of this

network is independent of the position of the discriminative feature $\ell$. Indeed, when the filter $\phi$ takes the optimal value $\phi[\ell] = 1/m[\ell]$ and $\phi[t] = 0$ for all $t \neq \ell$, the aliasing effect of $A$ can be neglected.

We study the loss landscape when optimizing a quadratic loss, $J(\boldsymbol{\theta}, \boldsymbol{\phi}; \boldsymbol{x}, y) = (y - f_{\boldsymbol{\theta},\boldsymbol{\phi}}(\boldsymbol{x}))^2$. In this setting, the following lemma describes the statistics of the geometry of the loss landscape.

**Lemma 1** (Average curvature of the loss landscape). *Assuming that the training parameters are distributed according to $\boldsymbol{\theta} \sim \mathcal{N}(\mathbf{0}, \sigma_{\boldsymbol{\theta}}^2 \boldsymbol{I}_M)$ and $\boldsymbol{\phi} \sim \mathcal{N}(\mathbf{0}, \sigma_{\boldsymbol{\phi}}^2 \boldsymbol{I}_D)$, the average weight Hessian of the loss with respect to $\boldsymbol{\phi}$ satisfies* $\mathbb{E}\, \nabla_{\boldsymbol{\phi}}^2 J(\boldsymbol{\theta}, \boldsymbol{\phi}; \boldsymbol{x}, y) = \underbrace{2\epsilon^2 \boldsymbol{m}^2[\ell] \sigma_{\boldsymbol{\theta}}^2 \operatorname{diag}(\boldsymbol{e}_\ell)}_{signal} + \underbrace{2\sigma^2 \sigma_{\boldsymbol{\theta}}^2 \operatorname{diag}(\boldsymbol{m}^2)}_{noise}.$

*Proof.* See Sec. B.2 of Supp. material[3]. $\qquad\square$

The curvature of the loss landscape can thus be decomposed into two terms: the curvature introduced by the discriminative signal, and the curvature introduced by the non-discriminative noise. A quantity that will control the speed of convergence of SGD will be the ratio between the curvature due to the signal component and the maximum curvature of the noise $\zeta(\ell) = \epsilon^2 \boldsymbol{m}^2[\ell] / \sigma^2 \max(\boldsymbol{m}^2)$ [29].

Intuitively, if $\zeta(\ell) \gg 1$, a small enough learning rate will quickly optimize the network in the direction of the optimal solution, avoiding big gradient oscillations caused by the non-discriminative components' curvature. On the contrary, if $\zeta(\ell) \lesssim 1$, the speed of convergence will be much slower due to the big oscillations introduced by the greatly curved noise components (see Fig. 4b and Fig. 2).

## 3  NAD computation

The choice of the Fourier basis so far was almost arbitrary, and there is no reason to suspect that it will capture the full directional inductive bias of all CNNs. NADs characterize this general bias, but it is clear that trying to identify them by measuring the performance of a neural network on many linearly separable datasets parameterized by a random $\boldsymbol{v}$ would be extremely inefficient[4]. It is therefore of paramount importance that we find another way to compute NADs without training.

In this sense, we will study the behaviour of a given architecture when it tries to solve a very simple discriminative task: classifying two data samples, $(\boldsymbol{x}, +1)$ and $(\boldsymbol{x} + \boldsymbol{v}, -1)$. We call them a *discriminative dipole*. Remarkably, studying this simple problem is enough to identify the NADs.

In general, given a discriminative dipole and a network $f_{\boldsymbol{\theta}} : \mathbb{R}^D \to \mathbb{R}$, parameterized by a general set of weights $\boldsymbol{\theta}$, we say that $f_{\boldsymbol{\theta}}$ has a high confidence in discriminating the dipole if it scores high on the metric $q_{\boldsymbol{\theta}}(\boldsymbol{v}) = g\left(|f_{\boldsymbol{\theta}}(\boldsymbol{x}) - f_{\boldsymbol{\theta}}(\boldsymbol{x} + \boldsymbol{v})|\right)$, where $g(t)$ can be any increasing function on $t \geq 0$, e.g., $g(t) = t^2$. In practice, we approximate $q_{\boldsymbol{\theta}}(\boldsymbol{v}) \approx g\left(|\boldsymbol{v}^T \nabla_{\boldsymbol{x}} f_{\boldsymbol{\theta}}(\boldsymbol{x})|\right)$ using a first-order Taylor expansion of $f_{\boldsymbol{\theta}}(\boldsymbol{x} + \boldsymbol{v})$ around $\boldsymbol{x}$.

As shown in Fig. 2, the directional bias can be identified based on the speed of convergence of a training algorithm. In the case of the dipole metric, this speed will depend on the size of $\|\nabla_{\boldsymbol{\theta}} q_{\boldsymbol{\theta}}(\boldsymbol{v})\|$. In expectation, this magnitude can be bounded by the following lemma.

**Lemma 2.** *Let $g$ be any increasing function on $t > 0$ with $|g'(t)| \leq \alpha|t| + \beta$, where $\alpha, \beta \geq 0$. Then*

$$\mathbb{E}_{\boldsymbol{\theta}} \|\nabla_{\boldsymbol{\theta}} q_{\boldsymbol{\theta}}(\boldsymbol{v})\| \leq \alpha^2 \sqrt{\mathbb{E}_{\boldsymbol{\theta}} |\boldsymbol{v}^T \nabla_{\boldsymbol{x}} f_{\boldsymbol{\theta}}(\boldsymbol{x})|^2} \sqrt{\mathbb{E}_{\boldsymbol{\theta}} \|\nabla_{\boldsymbol{\theta},\boldsymbol{x}}^2 f_{\boldsymbol{\theta}}(\boldsymbol{x}) \boldsymbol{v}\|^2} + \beta \mathbb{E}_{\boldsymbol{\theta}} \|\nabla_{\boldsymbol{\theta},\boldsymbol{x}}^2 f_{\boldsymbol{\theta}}(\boldsymbol{x}) \boldsymbol{v}\|. \quad (3)$$

*Proof.* See Sec. B.3 of Supp. material. $\qquad\square$

The right-hand side of (3) upper bounds $\mathbb{E}_{\boldsymbol{\theta}} \|\nabla_{\boldsymbol{\theta}} q_{\boldsymbol{\theta}}(\boldsymbol{v})\|$, and its magnitude with respect to $\boldsymbol{v}$ is controlled by the eigenvectors of $\mathbb{E}_{\boldsymbol{\theta}} \nabla_{\boldsymbol{x}} f_{\boldsymbol{\theta}}(\boldsymbol{x}) \nabla_{\boldsymbol{x}}^T f_{\boldsymbol{\theta}}(\boldsymbol{x})$ and the expected right singular vectors of $\nabla_{\boldsymbol{\theta},\boldsymbol{x}}^2 f_{\boldsymbol{\theta}}(\boldsymbol{x})$. We expect therefore that the NADs of an architecture are tightly linked to these vectors. The following example analyzes this relation for the deep linear network of Sec. 2.2.

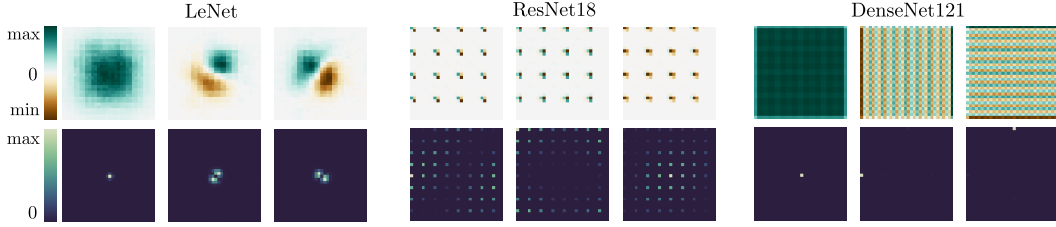

Figure 5: First three NADs of state-of-the-art CNNs in computer vision (more in Sec. D.1 of Supp. material). **Top row** shows NADs in pixel space and **bottom row** their energy in the Fourier domain.

**Example.** *Let $\phi \sim \mathcal{N}(\mathbf{0}, \sigma_\phi^2 \mathbf{I}_D)$ and $\boldsymbol{\theta} \sim \mathcal{N}(\mathbf{0}, \sigma_{\boldsymbol{\theta}}^2 \mathbf{I}_M)$. The covariance of the input gradient of the linear model of pooling $f_{\boldsymbol{\theta}, \phi}$ is*

$$\mathbb{E}_{\boldsymbol{\theta}, \phi} \nabla_{\boldsymbol{x}} f_{\boldsymbol{\theta}, \phi}(\boldsymbol{x}) \nabla_{\boldsymbol{x}}^T f_{\boldsymbol{\theta}, \phi}(\boldsymbol{x}) = \sigma_\phi^2 \sigma_{\boldsymbol{\theta}}^2 \operatorname{diag}\left(\boldsymbol{m}^2\right), \tag{4}$$

*and its eigenvectors are the canonical basis elements of $\mathbb{R}^D$, sorted by the entries of the prefilter $\boldsymbol{m}^2$. Surprisingly, the expected right singular vectors of $\nabla_{\boldsymbol{\theta}, \boldsymbol{x}}^2 f_{\boldsymbol{\theta}}(\boldsymbol{x})$ coincide with these eigenvectors,*

$$\mathbb{E}_{\boldsymbol{\theta}, \phi} \nabla_{(\boldsymbol{\theta}, \phi), \boldsymbol{x}}^2 f_{\boldsymbol{\theta}, \phi}(\boldsymbol{x})^T \nabla_{(\boldsymbol{\theta}, \phi), \boldsymbol{x}}^2 f_{\boldsymbol{\theta}, \phi}(\boldsymbol{x}) = \left( \sigma_{\boldsymbol{\theta}}^2 + \frac{\sigma_\phi^2}{S} \right) \operatorname{diag}(\boldsymbol{m}^2). \tag{5}$$

*Again, this result agrees with what has been shown in Sec. 2.2 where the NADs are also ordered according to the entries of $\boldsymbol{m}^2$.*

*Proof.* See Sec. C.1 of Supp. material. □

At this stage, it is important to highlight that these eigenvectors and singular vectors need not coincide, in general. However, as we will see in practice, these bases are surprisingly aligned for most networks, suggesting that the structure of these vectors is commonly rooted on some fundamental property of the architecture. Besides, although it could be argued that the bound in (3) is just an artefact of the choice of dipole metric, we provide below an alternative interpretation on the connection between $\mathbb{E}_{\boldsymbol{\theta}} \nabla_{\boldsymbol{x}} f_{\boldsymbol{\theta}}(\boldsymbol{x}) \nabla_{\boldsymbol{x}}^T f_{\boldsymbol{\theta}}(\boldsymbol{x})$, $\nabla_{\boldsymbol{\theta}, \boldsymbol{x}}^2 f_{\boldsymbol{\theta}}(\boldsymbol{x})$ and NADs.

On the one hand, $\nabla_{\boldsymbol{\theta}, \boldsymbol{x}}^2 f_{\boldsymbol{\theta}}(\boldsymbol{x})$ can be interpreted as a magnitude that controls the network tendency to create a decision boundary along a given direction, i.e., its right singular values quantify the inclination of a network to align $\nabla_{\boldsymbol{x}} f_{\boldsymbol{\theta}}(\boldsymbol{x})$ with a discriminative direction $\boldsymbol{v}$. On the other hand, the eigenvalues of $\mathbb{E}_{\boldsymbol{\theta}} \nabla_{\boldsymbol{x}} f_{\boldsymbol{\theta}}(\boldsymbol{x}) \nabla_{\boldsymbol{x}}^T f_{\boldsymbol{\theta}}(\boldsymbol{x})$ bound the *a priori* hardness to find a solution discriminating in a given direction. Specifically, considering the quadratic case, i.e., $g(t) = t^2$, we can estimate the volume of solutions that achieve a certain dipole metric $\eta$, $\mathbb{P}\left(q_{\boldsymbol{\theta}}(\boldsymbol{v}) \geq \eta\right)$. Indeed, an approximate bound to this volume using Markov's inequality depends only on the gradient covariance

$$\mathbb{P}\left(q_{\boldsymbol{\theta}}(\boldsymbol{v}) \geq \eta\right) \approx \mathbb{P}\left(\left(\boldsymbol{v}^T \nabla_{\boldsymbol{x}} f_{\boldsymbol{\theta}}(\boldsymbol{x})\right)^2 \geq \eta\right) \leq \frac{\boldsymbol{v}^T \left(\mathbb{E}_{\boldsymbol{\theta}} \nabla_{\boldsymbol{x}} f_{\boldsymbol{\theta}}(\boldsymbol{x}) \nabla_{\boldsymbol{x}}^T f_{\boldsymbol{\theta}}(\boldsymbol{x})\right) \boldsymbol{v}}{\eta}. \tag{6}$$

If the quadratic form is very low, the space of solutions achieving a certain $q_{\boldsymbol{\theta}}(\boldsymbol{v})$ is small. That is, it is much harder for a network to find solutions that optimize $q_{\boldsymbol{\theta}}(\boldsymbol{v})$ if the discriminative direction of the dipole is aligned with the eigenvectors associated to small eigenvalues of the gradient covariance.

Finally, note that the proposed eigendecomposition of the gradient covariance bares similarities with the techniques used to study neural networks in the mean-field regime [30–32]. These works study the effect of initialization and non-linearities on the Jacobian of inifinitely-wide networks to understand their trainability. In contrast, we analyze the properties of finite-size architectures and investigate the directionality of the singular vectors to explain the role of NADs in generalization. Analyzing the connections of NADs with these works will be subject of future research.

### 3.1 NADs of CNNs

For most deep networks, however, it is not tractable to analytically compute these decompositions in closed form. For this reason, we can apply Monte-Carlo sampling to estimate them. As we mentioned

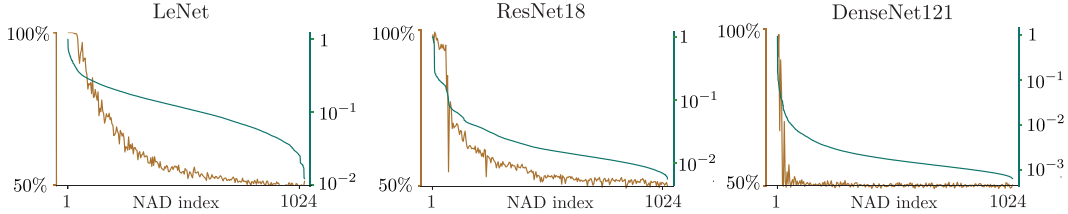

Figure 6: **(Green)** Normalized covariance eigenvalues and **(brown)** test accuracies of common state-of-the-art CNNs trained on linearly separable distributions parameterized by their NADs.

above, the bases derived from $\nabla^2_{\boldsymbol{\theta},\boldsymbol{x}} f_{\boldsymbol{\theta}}(\boldsymbol{x})$ and $\nabla_{\boldsymbol{x}} f_{\boldsymbol{\theta}}(\boldsymbol{x})$ are surprisingly very similar for most networks. Nevertheless, we observe that the approximation of NADs through the eigendecomposition of the gradient covariance is numerically more stable[5]. For this reason, we use these in the remainder of the paper.

Fig. 5 shows a few examples of NADs from several CNNs and illustrates their diversity. Focusing on these patterns, we can say that NADs act as a unique signature of an architecture. Remarkably, even though the supports of the energy in the Fourier domain of the first few NADs are included in the high accuracy regions of Fig. 1, not all NADs are sparse in the spectral domain. In particular, the NADs of a ResNet-18 look like combs of spikes in the Fourier domain. Similarly, the NADs do not follow a uniform ordering from low to high frequencies (cf. DenseNet-121 in Fig. 5). This suggests that each CNN relies on a unique set of features of training data to discriminate between different classes.

Analyzing how the exact construction of each architecture modulates the NADs of a network is out of the scope of this work. We believe that future research in this direction should focus on describing the individual contribution of different layers to understand the shape of NADs on CNNs.

## 4 NADs and generalization

In this section, we investigate the role of NADs on generalization in different scenarios. First, in the linearly separable setting, and later for CIFAR-10 dataset. In this sense, we will demonstrate that NADs are important quantities affecting the generalization properties of an architecture.

### 4.1 Learning linearly separable datasets

When a dataset is linearly separable by a single feature, the NADs completely characterize the performance of a network on this task. To show this, we replicate the experiments of Sec. 2, but this time using the NADs to parameterize the different distributions. In Fig. 6 we can see that the performance of these architectures monotonically decreases for higher NADs, and recall that in the Fourier basis (see Fig. 1), the performance did not monotonically decrease with frequency.

Observe as well that for the networks with lower rank of their gradient covariance (see Fig. 6), i.e., with a faster decay on its eigenvalues, the drop in accuracy happens at earlier indices and it is much more pronounced. In this sense, the ResNet-18 and DenseNet-121 that perform best on vision datasets such as ImageNet, ironically are the ones with the stronger bias on linearly separable datasets.

### 4.2 Learning CIFAR-10 dataset

Finally, we provide two experiments that illustrate the role of NADs beyond linearly separable cases.

**NADs define the order of selection of different discriminative features** First we borrow concepts from the data poisoning literature [33] as a way to probe the order in which features are selected by a given network. In particular, we do this by modifying all images in the CIFAR-10 training set to include a highly discriminative feature (carrier) aligned with a certain NAD. We repeat this experiment for multiple NADs and measure the test accuracy on the original CIFAR-10 test set.

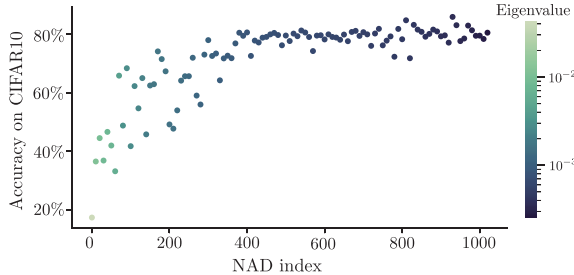

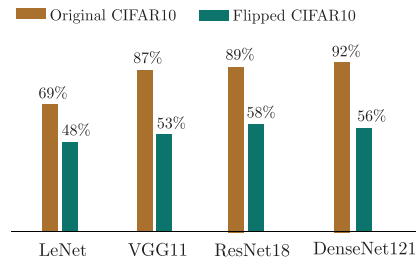

Figure 7: Accuracy on CIFAR-10 of a ResNet-18 when trained on multiple versions of poisoned data with a carrier ($\epsilon = 0.05$) at different NAD indices.

Figure 8: Test accuracies of state-of-the-art CNNs on the standard CIFAR-10 dataset and its flipped version.

An easy way to introduce a poisonous carrier on a sample is to substitute its content on a given direction by $\pm\epsilon$. CIFAR-10 has 10 classes and three color channels. Therefore, we can use two consecutive NADs applied on the different channels to encode a carrier that can poison this dataset. Note that, for any $\epsilon > 0$, this small modification on the training set renders the training set linearly separable using only the poisonous features. But, a classifier that only uses these features will not be able to generalize to the unpoisoned CIFAR-10 test set.

Fig. 7 shows the result of this experiment when the carriers are placed at the $i$th and $(i+1)$th NADs. For carriers placed at the first NADs the test accuracy is very low, showing that the network ignores most generalizing features. On the other hand, when the carrier is placed at the end of the sequence, where the extra feature is harder to learn (cf. Fig.6), the network can almost perfectly generalize.

A possible explanation of this behavior could be that during training, among all possible separating solutions, a network converges to the one that can discriminate the training data using the lowest NAD indices, i.e., using features spanned by the first NADs. In this sense, when a carrier is placed at a given NAD index, the network can only identify those generalizing features spanned by the NADs before the carrier, and ignores all those after it.

**NADs are necessary for generalization** To further support the previous explanation, we investigate the role of NADs as filters of discriminating solutions. In particular, we test the possible positive synergies arising from the alignment of NADs with the generalizing features of the training set. Specifically, we train multiple CNNs using the same hyperparameters on two representations of CIFAR-10: the original representation, and a new one in which we flip the representation of the data in the NAD basis. That is, for every sample $\boldsymbol{x}$ in the training and test sets we compute $\boldsymbol{x}' = \boldsymbol{U}\,\mathrm{flip}(\boldsymbol{U}^T\boldsymbol{x})$, where $\boldsymbol{U}$ represents a matrix with NAD vectors as its columns. Note that applying this transformation equates to a linear rotation of the input space and has no impact on the information of the data distribution. In fact, training on both representations yields approximately $0\%$ training error.

Fig. 8 shows the result of these experiments where we see that the performance of the networks trained on the flipped datasets is significantly lower than those on the original CIFAR-10. As demonstrated by the low accuracies on the flipped datasets, misaligning the inductive bias of these architectures with the datasets makes them prone to overfit to non-generalizing and spurious "noise". We see this effect as a supporting evidence that through the years the community has managed to impose the right inductive biases in deep neural architectures to classify the standard vision benchmarks.

## 5   Conclusion

In this paper we described a new type of model-driven inductive bias that controls generalization in deep neural networks: the directional inductive bias. We showed that this bias is encoded by an orthonormal set of vectors for each architecture, which we coined the NADs, and that these characterize the selection of discriminative features used by CNNs to separate a training set. In [12], researchers highlighted that a neural network memorizes a dataset when this has no discriminative information. In our work we complement this observation, and show that a network may prefer memorization over generalization, even when there exists a highly discriminative feature in the

dataset. Surprisingly, this phenomenon is not only attributable to some property of the data, but also to the structure of the architecture.

Future research should focus on providing a better theoretical understanding of the mechanisms that determine the NADs, but also on describing their role on the dynamics of training. Extending the NAD discovery algorithms to other families of architectures like graph neural networks [21] or transformers [20] would be a natural next step. All in all, we believe that our findings can have potential impacts on future research in designing better architectures and AutoML [34], paving the way for better aligning the inductive biases of deep networks with *a priori* structures on real data.

Finally, it is important to note that our results mostly apply to cases in which the data was fully separable, i.e. there was no label noise. And even more specifically, to the linearly separable case. In this sense, it still remains an open problem to understand how the directional inductive bias of deep learning influences neural networks trying to learn non-separable datasets.

## Broader Impact

In this work we reveal the directional inductive bias of deep learning and describe its role in controlling the type of functions that neural networks can learn. The algorithm that we introduced to characterize it can help understand the reasons for the success or alternatively the modes of failure of most modern CNNs. Our work is mainly fundamental in the sense that it is not geared towards an application, but theory always has some downstream implications on the society as enabler of future applications.

We see potential applications of our work on AutoML [34] as the main positive impact of our research. In particular, we believe that incorporating prior knowledge into the neural architecture search loop [35] can cut most computational and environmental costs of this procedure. Specifically, with the current trend in deep learning towards building bigger and computationally greedier models [36], the impact of machine learning on the environment is becoming a pressing issue [37]. Meanwhile, this trend is raising the bar on the needed resources to use these models and research in deep learning is getting concentrated around a few big actors. In this sense, we believe that gaining a better understanding of our current models will be key in circumventing the heavy heuristics necessary to deploy deep learning today, thus enabling the democratization of this technology [38].

On the other hand, we see the main possible negative implication of our work in the malicious use of NADs to boost the adversarial capacity of new evasion/backdoor attacks [39]. This could potentially exploit the sensitivity of neural networks to NADs to generate more sophisticated adversarial techniques. Machine learning engineers should be aware of such vulnerabilities when designing new architectures, especially for safety-critical applications.

## Acknowledgments

We thank Maksym Andriushchenko, Hadi Daneshmand, and Clément Vignac, for their fruitful discussions and feedback. This work has been partially supported by the CHIST-ERA program under Swiss NSF Grant 20CH21_180444, and partially by Google via a Postdoctoral Fellowship and a GCP Research Credit Award.

## Footnotes

[1]The exact settings of this experiment will be described in Sec. 2. In general, all training and evaluation setups, hyperparameters, number of training samples, and network performances are listed in the Supp. material.

[2]For the exact procedure and more experiments with similar findings see Sec. A.1 and A.2 of Supp. material.

[3]A similar result for $\boldsymbol{\theta}$ can be found in Sec. B.2 of Supp. material.

[4]For most of these datasets the network outputs the same performance as seen in Sec. A.3 of Supp. material.

[5]A detail description of the algorithmic implementation used to approximate NADs and examples of more NADs computed using both decompositions are given in Sec. D.1 and Sec. D.2 of Supp. material.

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
