[Supplementary Material]

# Supplementary material: Neural Anisotropy Directions

# Contents

## A  Experiments on linearly separable datasets

### A.1  General training setup

Regarding the construction of the synthetic datasets used for the experiments of Sec. 2 and Sec. 3.1, recall that $\mathcal{D}(\boldsymbol{v})$ is a linearly separable distribution parameterized by a unit vector $\boldsymbol{v} \in \mathbb{S}^{D-1}$, such that any sample $(\boldsymbol{x}, y) \sim \mathcal{D}(\boldsymbol{v})$ satisfies $\boldsymbol{x} = \epsilon y \boldsymbol{v} + \boldsymbol{w}$, with noise $\boldsymbol{w} \sim \mathcal{N}\left(\boldsymbol{0}, \sigma^2(\boldsymbol{I}_D - \boldsymbol{v}\boldsymbol{v}^T)\right)$ orthogonal to the direction $\boldsymbol{v}$, and with label $y$ sampled from $\{-1, +1\}$ with equal probability. An illustration of such dataset is shown in Fig. S1.

Figure S1: Schematic of the parameters of $\mathcal{D}(\boldsymbol{v})$.

In general, the generated synthetic data correspond to $32 \times 32$ grayscale images, with the standard settings being $10,000$ training samples, $10,000$ test samples, and $\epsilon = 1$. The value of $\sigma$ varies depending on the experiment under study.

Regarding the setup and parameters for training the networks used for the experiments of Sec. 2 and Sec. 3.1: they were all trained for 20 epochs, on batches of size 128, minimizing a Cross-Entropy loss using SGD with a linearly decaying learning rate (max lr. 0.5) and without any explicit regularization.

At this point let us note that we did not perform any extensive hyperparameter tuning to arrive at this configuration. In fact, we empirically observed that such parameters were good enough to reveal the quantities of interest (i.e., directional inductive bias) and did not tune them further. In general, all of our observations are relative in the sense that we do not focus on the exact values, e.g., test accuracy or training iterations (except reaching almost zero training loss), but their relative differences for different distributions.

### A.2  Experiments on DFT basis

#### A.2.1  Basis generation

Recall that the DFT $\mathcal{F} : \mathbb{C}^D \to \mathbb{C}^D$ is a complex linear operator acting in the complex plane. For this reason, the basis obtained from transforming the canonical basis through the DFT, i.e., $\boldsymbol{v}_i = \mathcal{F}(\boldsymbol{e}_i)$ is a complex basis. In this work we are interested in dealing with real signals, and as such we need to modify this basis such that it is an orthonormal basis of the real space $\mathbb{R}^D$.

We can do that by leveraging the conjugate symmetry of the DFT of real signals. Let $\boldsymbol{x} \in \mathbb{R}^D$ with Fourier transform $\hat{\boldsymbol{x}} = \mathcal{F}(\boldsymbol{x}) \in \mathbb{C}^D$. Then,

$$\hat{\boldsymbol{x}}[\![t]\!]_D = \hat{\boldsymbol{x}}^\star[\![-t]\!]_D,$$

where $\hat{\boldsymbol{x}}^\star$ represents the complex conjugate of $\hat{\boldsymbol{x}}$. This means that, for real signals, half of the DFT is redundant, and one can use only $\lfloor D/2 \rfloor + 1$ complex coefficients to represent a real signal. We are interested in obtaining a basis of $\mathbb{R}^D$ which is sparse in the Fourier domain. However, note that for any index $t$, $\langle \mathcal{F}^{-1}(\boldsymbol{e}_t), \mathcal{F}^{-1}(j\boldsymbol{e}_t) \rangle = 0$, with $j = \sqrt{-1}$. For this reason, we can create a basis of $\mathbb{R}^D$ using $\lfloor D/2 \rfloor + 1$ real coefficients and $\lfloor D/2 \rfloor$ imaginary coefficients by exploiting their conjugate

symmetries, i.e.,

$$\boldsymbol{v}_i^{\mathrm{Re}} = \frac{1}{\sqrt{2}} \mathcal{F}^{-1} \left( \boldsymbol{e}_{[\![i]\!]_D} + \boldsymbol{e}_{[\![-i]\!]_D} \right) \qquad\qquad i = 0, \dots, \lfloor D/2 \rfloor$$

$$\boldsymbol{v}_i^{\mathrm{Im}} = \frac{1}{\sqrt{2}} \mathcal{F}^{-1} \left( j\boldsymbol{e}_{[\![i]\!]_D} - j\boldsymbol{e}_{[\![-i]\!]_D} \right) \qquad\qquad i = 1, \dots, \lfloor D/2 \rfloor$$

Fortunately, most numerical linear algebra libraries avoid the need to keep track of these symmetries and include some routine to directly compute the Fourier transform and its inverse on real signals (RFFT). This is especially useful on bidimensional signals, like images, where the RFFT of a signal has $D \times \lfloor D/2 \rfloor + 1$ complex coefficients. Nevertheless, despite the redundancies, it is a common convention in the image processing community to plot the full Fourier spectrum of an image including positive and negative frequencies (indices). In our plots, we follow this convention, and artificially create the symmetries on the negative indices to ease readability[1].

All the results that we have shown so far using the DFT basis show only the results for the directions obtained from manipulating the real coefficients. Nevertheless, the results do not change in nature when one repeats them on the imaginary elements as well. We provide Fig. S2 as a validation of this, where we repeated the same experiment as in Fig. 1 but using the directions parameterized by the imaginary coefficients, i.e., $\boldsymbol{v}_i^{\mathrm{Im}}$. Note that, because the number of basis vectors parameterized by the imaginary coefficients is smaller, there are four gaps in Fig. S2. These are just artefacts of the visualization, as these distributions do not exist in reality.

Figure S2: Imaginary part of DFT

## A.2.2 Different noise levels

Fig. S3 illustrates the test accuracies of various architectures under different noise levels $\sigma$. Regardless the noise level, a logistic regression can always perfectly generalize to the test data. On the contrary, LeNet seems to fail to generalize to a few distributions even in the absence of noise, while the noisier the data the more its performance degrades. The other architectures exhibit similar behaviour: they properly generalize when there is no noise, while their performance drops as the noise level increases. Finally, note that ResNet-18 seems to be slightly more robust to noise compared to the other CNNs (cf. Fig. S3b with $\sigma = 1$).

## A.3 Experiments on random basis

As mentioned in Sec. 3, trying to identify the NADs of an architecture by measuring its performance on many linearly separable datasets parameterized by a random direction $\boldsymbol{v}$, would be extremely inefficient. To demonstrate this, we repeat the same experiment performed in Sec. 2, but instead of constructing the training sets $\mathcal{D}(\boldsymbol{v})$ using $\boldsymbol{v}$s taken from the 2D-DCT basis elements, each $\boldsymbol{v}$ now corresponds to a basis element of a random orthonormal matrix $\boldsymbol{U} \in \mathrm{SO}(D)$.

The results of this experiment are illustrated in Fig. S4. Indeed, it is clear that such procedure will never be able to reveal the directional inductive bias of an architecture: for most of the datasets the networks output the same performance, thus it is impossible to interpret if these directions are aligned with the directional inductive bias of the architecture under study.

(a) Test accuracies for $\sigma = 0$.

(b) Test accuracies for $\sigma = 1$.

(c) Test accuracies for $\sigma = 3$.

Figure S3: Test accuracies using different training sets drawn from $\mathcal{D}(\boldsymbol{v})$ ($\epsilon = 1$, with $10{,}000$ training samples and $10{,}000$ test samples) for different levels of $\sigma$. Directions $\boldsymbol{v}$ taken from the basis elements of the 2D-DFT. Each pixel corresponds to a linearly separable dataset.

(a) LeNet

(b) ResNet-18

Figure S4: Test accuracy of two CNNs trained using different training sets drawn from $\mathcal{D}(\boldsymbol{v})$ ($\epsilon = 1$, and $\sigma = 3$) with orthogonal random $\boldsymbol{v}$.

## B  Deferred proofs

### B.1  Proof of Theorem 1

We give here the proof of Theorem 1 stating the Bayes optimal classification accuracy achieved on a linearly separable distribution transformed through a linear pooling layer. We restate the theorem to ease readability.

**Theorem** (Bayes optimal classification accuracy after pooling). *The best achievable accuracy on the distribution of $(\boldsymbol{z}, y)$ can be written as*

$$1 - \mathcal{Q}\left(\frac{\epsilon}{2\sigma}\gamma(\ell)\right) \quad with \quad \gamma^2(\ell) = \frac{S|\hat{\boldsymbol{m}}[\ell]|^2}{\sum_{k=1}^{S-1}|\hat{\boldsymbol{m}}[\![\ell + k \cdot M]\!]_D|^2},$$

*and $\mathcal{Q}(\cdot)$ representing the tail distribution function of the standard normal distribution.*

*Proof.* Without loss of generality, let $(\boldsymbol{x}, y)$ be a random sample with $y \sim \mathcal{U}\{-1, 1\}$ and whose Fourier transform satisfies

$$\hat{\boldsymbol{x}} = \epsilon y \boldsymbol{e}_\ell + \hat{\boldsymbol{w}} \qquad with \qquad \hat{\boldsymbol{w}} \sim \mathcal{CN}(\boldsymbol{0}, \mathrm{diag}(\boldsymbol{\sigma}^2)),$$

where $\mathcal{CN}(\boldsymbol{0}, \mathrm{diag}(\boldsymbol{\sigma})^2)$ denotes a circularly symmetric complex Gaussian distribution with complex covariance $\mathrm{diag}(\boldsymbol{\sigma}^2)$.

Because all entries of $\hat{\boldsymbol{x}}$ are uncorrelated, the best accuracy on the distribution of $(\boldsymbol{x}, y)$, $\alpha_{\mathrm{opt}}$, would be the same as that of the distribution of $(\mathfrak{R}(\hat{\boldsymbol{x}}[\ell]), y)$, i.e., $\mathfrak{R}(\hat{\boldsymbol{x}}[\ell])|y = +1 \sim \mathcal{N}(\epsilon, \boldsymbol{\sigma}^2[\ell]/2)$ and $\mathfrak{R}(\hat{\boldsymbol{x}}[\ell])|y = -1 \sim \mathcal{N}(-\epsilon, \boldsymbol{\sigma}^2[\ell]/2)$. Hence,

$$\alpha_{\mathrm{opt}} = 1 - \mathcal{Q}\left(\frac{\sqrt{2}\epsilon}{2\boldsymbol{\sigma}[\ell]}\right).$$

Nevertheless, we are interested on the accuracy on the distribution of $(\boldsymbol{z}, y)$, when $(\boldsymbol{x}, y) \sim \mathcal{D}(\boldsymbol{v}_\ell)$ with $\boldsymbol{v}_\ell = \mathcal{F}(\boldsymbol{e}_\ell)$, whose spectrum satisfies

$$\hat{\boldsymbol{z}} = \epsilon y \hat{\boldsymbol{m}}[\ell] \boldsymbol{e}'_{[\![\ell]\!]_M} + \mathrm{diag}(\hat{\boldsymbol{m}})\hat{\boldsymbol{w}}$$

with $\boldsymbol{e}'_{[\![\ell]\!]_M} \in \mathbb{C}^M$ the $(\ell \bmod M)$th canonical basis vector of $\mathbb{R}^M$, and $\mathrm{diag}(\hat{\boldsymbol{m}})\hat{\boldsymbol{w}} \sim \mathcal{CN}(\boldsymbol{0}, \mathrm{diag}(\boldsymbol{\xi}))$ with

$$\boldsymbol{\xi}^2[\![\ell]\!]_M = \frac{\sigma^2}{S}\sum_{k=1}^{S-1}|\hat{\boldsymbol{m}}[\![\ell + k \cdot M]\!]_D|^2.$$

Again, the only signal component is at $\hat{\boldsymbol{z}}[\![\ell]\!]_M$. Hence, if we write

$$\gamma^2(\ell) = \frac{S|\hat{\boldsymbol{m}}[\ell]|^2}{\sum_{k=1}^{S-1}|\hat{\boldsymbol{m}}[\![\ell + k \cdot M]\!]_D|^2},$$

and finally the accuracy of the Bayes optimal classifier on the distribution of $(\boldsymbol{z}, y)$ can be explicitly described by

$$\alpha(\ell) = 1 - \mathcal{Q}\left(\frac{\sqrt{2}\epsilon|\hat{\boldsymbol{m}}[\ell]|}{2\boldsymbol{\xi}[\![\ell]\!]_M}\right) = 1 - \mathcal{Q}\left(\frac{\sqrt{2}\epsilon}{2\sigma}\gamma(\ell)\right).$$

$\square$

### B.2  Proof of Lemma 1

We detail here the proof of Lemma 1 describing the average curvature of the loss landscape for the deep linear network $f_{\boldsymbol{\theta}, \boldsymbol{\phi}}(\boldsymbol{x}) = \boldsymbol{\theta}^T \boldsymbol{A}(\boldsymbol{m} \odot \boldsymbol{\phi} \odot \boldsymbol{x})$ when optimizing the quadratic loss $J(\boldsymbol{\theta}, \boldsymbol{\phi}; \boldsymbol{x}, y) = (y - f_{\boldsymbol{\theta}, \boldsymbol{\phi}}(\boldsymbol{x}))^2$.

**Lemma** (Average curvature of the loss landscape). *Assuming that the training parameters are distributed according to $\boldsymbol{\theta} \sim \mathcal{N}(\boldsymbol{0}, \sigma_{\boldsymbol{\theta}}^2 \boldsymbol{I}_M)$ and $\boldsymbol{\phi} \sim \mathcal{N}(\boldsymbol{0}, \sigma_{\boldsymbol{\phi}}^2 \boldsymbol{I}_D)$, the average weight Hessian of the loss with respect to $\boldsymbol{\phi}$ satisfies $\mathbb{E}\,\nabla_{\boldsymbol{\phi}}^2 J(\boldsymbol{\theta}, \boldsymbol{\phi}; \boldsymbol{x}, y) = 2\epsilon^2 \boldsymbol{m}^2[\ell]\sigma_{\boldsymbol{\theta}}^2 \mathrm{diag}(\boldsymbol{e}_\ell) + 2\sigma^2\sigma_{\boldsymbol{\theta}}^2 \mathrm{diag}(\boldsymbol{m}^2)$.*

*Proof.* Let us start by computing the gradients for a generic loss $J(\boldsymbol{\theta}, \boldsymbol{\phi}; \boldsymbol{x}, y) = q(z, y)$ with $z = f_{\boldsymbol{\theta}, \boldsymbol{\phi}(\boldsymbol{x})}$

$$\nabla_{\boldsymbol{\theta}} J(\boldsymbol{\theta}, \boldsymbol{\phi}; \boldsymbol{x}, y) = q'(z, y) \boldsymbol{A}(\boldsymbol{\phi} \odot \boldsymbol{m} \odot \boldsymbol{x})$$
$$\nabla_{\boldsymbol{\phi}} J(\boldsymbol{\theta}, \boldsymbol{\phi}; \boldsymbol{x}, y) = q'(z, y) (\boldsymbol{A}^T \boldsymbol{\theta}) \odot (\boldsymbol{m} \odot \boldsymbol{x}).$$

Therefore, the second derivatives are

$$\nabla_{\boldsymbol{\theta}}^2 J(\boldsymbol{\theta}, \boldsymbol{\phi}; \boldsymbol{x}, y) = q''(z, y) \boldsymbol{A}(\boldsymbol{\phi} \odot \boldsymbol{m} \odot \boldsymbol{x})(\boldsymbol{A}(\boldsymbol{\phi} \odot \boldsymbol{m} \odot \boldsymbol{x}))^T$$
$$\nabla_{\boldsymbol{\phi}}^2 J(\boldsymbol{\theta}, \boldsymbol{\phi}; \boldsymbol{x}, y) = q''(z, y) (\boldsymbol{A}^T \boldsymbol{\theta}) \odot (\boldsymbol{m} \odot \boldsymbol{x})((\boldsymbol{A}^T \boldsymbol{\theta}) \odot (\boldsymbol{m} \odot \boldsymbol{x}))^T$$
$$\nabla_{\boldsymbol{\theta}, \boldsymbol{\phi}}^2 J(\boldsymbol{\theta}, \boldsymbol{\phi}; \boldsymbol{x}, y) = q''(z, y)((\boldsymbol{A}^T \boldsymbol{\theta}) \odot (\boldsymbol{m} \odot \boldsymbol{x}))(\boldsymbol{A}(\boldsymbol{\phi} \odot \boldsymbol{m} \odot \boldsymbol{x}))^T +$$
$$+ q'(z, y) \boldsymbol{A} \operatorname{diag}(\boldsymbol{x} \odot \boldsymbol{m})$$
$$\nabla_{\boldsymbol{\phi}, \boldsymbol{\theta}}^2 J(\boldsymbol{\theta}, \boldsymbol{\phi}; \boldsymbol{x}, y) = q''(z, y) \boldsymbol{A}(\boldsymbol{\phi} \odot \boldsymbol{m} \odot \boldsymbol{x})((\boldsymbol{A}^T \boldsymbol{\theta}) \odot (\boldsymbol{m} \odot \boldsymbol{x}))^T +$$
$$+ q'(z, y) \operatorname{diag}(\boldsymbol{x} \odot \boldsymbol{m}) \boldsymbol{A}^T$$

Hence, the Hessian

$$\nabla^2 J(\boldsymbol{\theta}, \boldsymbol{\phi}; \boldsymbol{x}, y) = q''(z, y) \begin{bmatrix} \boldsymbol{A}(\boldsymbol{\phi} \odot \boldsymbol{m} \odot \boldsymbol{x}) \\ (\boldsymbol{A}^T \boldsymbol{\theta}) \odot (\boldsymbol{m} \odot \boldsymbol{x}) \end{bmatrix} \begin{bmatrix} \boldsymbol{A}(\boldsymbol{\phi} \odot \boldsymbol{m} \odot \boldsymbol{x}) \\ (\boldsymbol{A}^T \boldsymbol{\theta}) \odot (\boldsymbol{m} \odot \boldsymbol{x}) \end{bmatrix}^T +$$
$$+ q'(z, y) \begin{bmatrix} \boldsymbol{0} & \boldsymbol{A} \operatorname{diag}(\boldsymbol{x} \odot \boldsymbol{m}) \\ \operatorname{diag}(\boldsymbol{x} \odot \boldsymbol{m}) \boldsymbol{A}^T & \boldsymbol{0} \end{bmatrix}$$
$$= q''(z, y) \begin{bmatrix} \boldsymbol{A} \operatorname{diag}(\boldsymbol{\phi} \odot \boldsymbol{m}) \\ \operatorname{diag}(\boldsymbol{A}^T \boldsymbol{\theta} \odot \boldsymbol{m}) \end{bmatrix} \boldsymbol{x} \boldsymbol{x}^T \begin{bmatrix} \boldsymbol{A} \operatorname{diag}(\boldsymbol{\phi} \odot \boldsymbol{m}) \\ \operatorname{diag}(\boldsymbol{A}^T \boldsymbol{\theta} \odot \boldsymbol{m}) \end{bmatrix}^T +$$
$$+ q'(z, y) \begin{bmatrix} \boldsymbol{0} & \boldsymbol{A} \operatorname{diag}(\boldsymbol{x} \odot \boldsymbol{m}) \\ \operatorname{diag}(\boldsymbol{x} \odot \boldsymbol{m}) \boldsymbol{A}^T & \boldsymbol{0} \end{bmatrix}$$

When we optimize a square loss, $q''(z, y) = 2$ and $q'(z, y) = 2(z - y)$. Thus,

$$\nabla^2 J(\boldsymbol{\theta}, \boldsymbol{\phi}; \boldsymbol{x}, y) = 2 \begin{bmatrix} \boldsymbol{A} \operatorname{diag}(\boldsymbol{\phi} \odot \boldsymbol{m}) \\ \operatorname{diag}(\boldsymbol{A}^T \boldsymbol{\theta} \odot \boldsymbol{m}) \end{bmatrix} \boldsymbol{x} \boldsymbol{x}^T \begin{bmatrix} \boldsymbol{A} \operatorname{diag}(\boldsymbol{\phi} \odot \boldsymbol{m}) \\ \operatorname{diag}(\boldsymbol{A}^T \boldsymbol{\theta} \odot \boldsymbol{m}) \end{bmatrix}^T$$
$$+ \underbrace{2(z - y) \begin{bmatrix} \boldsymbol{0} & \boldsymbol{A} \operatorname{diag}(\boldsymbol{x} \odot \boldsymbol{m}) \\ \operatorname{diag}(\boldsymbol{x} \odot \boldsymbol{m}) \boldsymbol{A}^T & \boldsymbol{0} \end{bmatrix}}_{\boldsymbol{R}}.$$

Let $\boldsymbol{e}_\ell \in \mathbb{R}^D$ and $\boldsymbol{e}'_\ell \in \mathbb{R}^M$ be the $\ell$th canonical basis vectors of $\mathbb{R}^D$ and $\mathbb{R}^M$, respectively. Taking the expectation over the data we get

$$\mathbb{E}_{(\boldsymbol{x}, y)} \nabla^2 J(\boldsymbol{\theta}, \boldsymbol{\phi}; \boldsymbol{x}, y) = 2 \begin{bmatrix} \boldsymbol{A} \operatorname{diag}(\boldsymbol{\phi} \odot \boldsymbol{m}) \\ \operatorname{diag}(\boldsymbol{A}^T \boldsymbol{\theta} \odot \boldsymbol{m}) \end{bmatrix} (\epsilon^2 \operatorname{diag}(\boldsymbol{e}_\ell) + \sigma^2 \boldsymbol{I}_D) \begin{bmatrix} \boldsymbol{A} \operatorname{diag}(\boldsymbol{\phi} \odot \boldsymbol{m}) \\ \operatorname{diag}(\boldsymbol{A}^T \boldsymbol{\theta} \odot \boldsymbol{m}) \end{bmatrix}^T$$
$$+ \mathbb{E}_{(\boldsymbol{x}, y)} \boldsymbol{R}.$$

Here, the first summand can be decomposed in a signal and a noise component. The signal component is

$$\boldsymbol{S} = \begin{bmatrix} \boldsymbol{A} \operatorname{diag}(\boldsymbol{\phi} \odot \boldsymbol{m}) \\ \operatorname{diag}(\boldsymbol{A}^T \boldsymbol{\theta} \odot \boldsymbol{m}) \end{bmatrix} \epsilon^2 \operatorname{diag}(\boldsymbol{e}_\ell) \begin{bmatrix} \boldsymbol{A} \operatorname{diag}(\boldsymbol{\phi} \odot \boldsymbol{m}) \\ \operatorname{diag}(\boldsymbol{A}^T \boldsymbol{\theta} \odot \boldsymbol{m}) \end{bmatrix}^T$$
$$= \epsilon^2 \begin{bmatrix} \boldsymbol{\phi}[\ell] \boldsymbol{m}[\ell] \operatorname{diag}\left(\boldsymbol{e}'_{[\![\ell]\!]_M}\right) \\ \boldsymbol{\theta}[\![\ell]\!]_M \boldsymbol{m}[\ell] \operatorname{diag}\left(\boldsymbol{e}_\ell\right) \end{bmatrix} \begin{bmatrix} \boldsymbol{A} \operatorname{diag}(\boldsymbol{\phi} \odot \boldsymbol{m}) \\ \operatorname{diag}(\boldsymbol{A}^T \boldsymbol{\theta} \odot \boldsymbol{m}) \end{bmatrix}^T =$$
$$= \epsilon^2 \boldsymbol{m}^2[\ell] \begin{bmatrix} \boldsymbol{\phi}^2[\ell] \operatorname{diag}\left(\boldsymbol{e}'_{[\![\ell]\!]_M}\right) & \boldsymbol{\theta}[\![\ell]\!]_M \boldsymbol{\phi}[\ell] \boldsymbol{e}'_{[\![\ell]\!]_M} \boldsymbol{e}_\ell^T \\ \boldsymbol{\theta}[\![\ell]\!]_M \boldsymbol{\phi}[\ell] \boldsymbol{e}_\ell \boldsymbol{e}_{[\![\ell]\!]_M}^{\prime T} & \boldsymbol{\theta}^2[\![\ell]\!]_M \operatorname{diag}\left(\boldsymbol{e}_\ell\right) \end{bmatrix}.$$

122 The noise component is

$$
\boldsymbol{W} = \sigma^2 \begin{bmatrix} \boldsymbol{A}\operatorname{diag}(\boldsymbol{\phi}\odot\boldsymbol{m}) \\ \operatorname{diag}(\boldsymbol{A}^T\boldsymbol{\theta}\odot\boldsymbol{m}) \end{bmatrix} \begin{bmatrix} \boldsymbol{A}\operatorname{diag}(\boldsymbol{\phi}\odot\boldsymbol{m}) \\ \operatorname{diag}(\boldsymbol{A}^T\boldsymbol{\theta}\odot\boldsymbol{m}) \end{bmatrix}^T =
$$

$$
= \sigma^2 \begin{bmatrix} \boldsymbol{A}\operatorname{diag}(\boldsymbol{\phi}^2\odot\boldsymbol{m}^2) & \boldsymbol{A}\operatorname{diag}(\boldsymbol{\phi}\odot\boldsymbol{m})(\operatorname{diag}(\boldsymbol{A}^T\boldsymbol{\theta}\odot\boldsymbol{m}))^T \\ (\operatorname{diag}(\boldsymbol{A}^T\boldsymbol{\theta}\odot\boldsymbol{m}))(\boldsymbol{A}\operatorname{diag}(\boldsymbol{\phi}\odot\boldsymbol{m}))^T & \operatorname{diag}(\boldsymbol{A}^T\boldsymbol{\theta}^2\odot\boldsymbol{m}^2) \end{bmatrix}
$$

123 Taking the expectation over the parameters

$$
\mathbb{E}_{\boldsymbol{\theta},\boldsymbol{\phi}}\boldsymbol{S} = \epsilon^2\boldsymbol{m}^2[\ell]\begin{bmatrix} \sigma_{\boldsymbol{\phi}}^2\operatorname{diag}\left(\boldsymbol{e}'_{[\![\ell]\!]_M}\right) & \boldsymbol{0} \\ \boldsymbol{0} & \sigma_{\boldsymbol{\theta}}^2\operatorname{diag}\left(\boldsymbol{e}_\ell\right) \end{bmatrix}
$$

$$
\mathbb{E}_{\boldsymbol{\theta},\boldsymbol{\phi}}\boldsymbol{W} = \sigma^2\begin{bmatrix} \boldsymbol{A}\sigma_{\boldsymbol{\phi}}^2\operatorname{diag}(\boldsymbol{m}^2) & \boldsymbol{0} \\ \boldsymbol{0} & \sigma_{\boldsymbol{\theta}}^2\operatorname{diag}(\boldsymbol{m}^2) \end{bmatrix},
$$

124 and because $\mathbb{E}_{\boldsymbol{\theta}}z = \mathbb{E}_{\boldsymbol{\phi}}z = 0$, and $\mathbb{E}y = 0$, then $\mathbb{E}\boldsymbol{R} = \boldsymbol{0}$.

125 Overall, we see that

$$
\mathbb{E}\nabla^2 J(\boldsymbol{\theta},\boldsymbol{\phi};\boldsymbol{x},y) = \begin{bmatrix} \boldsymbol{H}_{\boldsymbol{\phi}} & \boldsymbol{0} \\ \boldsymbol{0} & \boldsymbol{H}_{\boldsymbol{\theta}} \end{bmatrix}
$$

126 with

$$
\boldsymbol{H}_{\boldsymbol{\phi}} = 2\epsilon^2\boldsymbol{m}^2[\ell]\sigma_{\boldsymbol{\theta}}^2\operatorname{diag}\left(\boldsymbol{e}_\ell\right) + 2\sigma^2\sigma_{\boldsymbol{\theta}}^2\operatorname{diag}(\boldsymbol{m}^2)
$$

$$
\boldsymbol{H}_{\boldsymbol{\theta}} = 2\epsilon^2\boldsymbol{m}^2[\ell]\sigma_{\boldsymbol{\phi}}^2\operatorname{diag}\left(\boldsymbol{e}'_{[\![\ell]\!]_M}\right) + 2\sigma^2\sigma_{\boldsymbol{\phi}}^2\boldsymbol{A}\operatorname{diag}(\boldsymbol{m}^2)
$$

127 □

## B.3 Proof of Lemma 2

129 We prove Lemma 2 under a slightly more general setting than in the text.

130 **Lemma.** *Let $g : [0,\infty] \to [0,\infty]$ be an increasing function with polynomially bounded first-order*
131 *derivative, i.e., $|g'(t)| \le \omega_n(|t|)$, where $\omega_n : \mathbb{R} \to \mathbb{R}$ is an $n$-order polynomial.*

132 *The expected value of $\|\nabla_{\boldsymbol{\theta}}q_{\boldsymbol{\theta}}(\boldsymbol{v})\|$ is bounded by*

$$
\mathbb{E}\|\nabla_{\boldsymbol{\theta}}q_{\boldsymbol{\theta}}(\boldsymbol{v})\| \le \sqrt{\mathbb{E}\omega_n^2\left(|\boldsymbol{v}^T\nabla_{\boldsymbol{x}}f_{\boldsymbol{\theta}}(\boldsymbol{x})|\right)}\sqrt{\mathbb{E}\|\nabla_{\boldsymbol{\theta},\boldsymbol{x}}^2 f_{\boldsymbol{\theta}}(\boldsymbol{x})\boldsymbol{v}\|^2}
$$

133 *Proof.* Using the polynomial bound on the derivative of $g$ and using Cauchy-Schwarz inequality we
134 can bound the expected norm of $\nabla_{\boldsymbol{\theta}}q_{\boldsymbol{\theta}}(\boldsymbol{x})$ as

$$
\begin{aligned}
\mathbb{E}\|\nabla_{\boldsymbol{\theta}}q_{\boldsymbol{\theta}}(\boldsymbol{v})\| &= \mathbb{E}|g'\left(|\boldsymbol{v}^T\nabla_{\boldsymbol{x}}f_{\boldsymbol{\theta}}(\boldsymbol{x})|\right)|\,\|\nabla_{\boldsymbol{\theta},\boldsymbol{x}}^2 f_{\boldsymbol{\theta}}(\boldsymbol{x})\boldsymbol{v}\| \\
&\le \mathbb{E}\omega_n\left(|\boldsymbol{v}^T\nabla_{\boldsymbol{x}}f_{\boldsymbol{\theta}}(\boldsymbol{x})|\right)\|\nabla_{\boldsymbol{\theta},\boldsymbol{x}}^2 f_{\boldsymbol{\theta}}(\boldsymbol{x})\boldsymbol{v}\| \\
&\le \sqrt{\mathbb{E}\omega_n^2\left(|\boldsymbol{v}^T\nabla_{\boldsymbol{x}}f_{\boldsymbol{\theta}}(\boldsymbol{x})|\right)}\sqrt{\mathbb{E}\|\nabla_{\boldsymbol{\theta},\boldsymbol{x}}^2 f_{\boldsymbol{\theta}}(\boldsymbol{x})\boldsymbol{v}\|^2}
\end{aligned}
$$

135 □

136 We see that this bound depends on the spectral decomposition of the moments of $\nabla_{\boldsymbol{x}}f_{\boldsymbol{\theta}}(\boldsymbol{x})$ up to
137 order $2n$, e.g., its covariance $\mathbb{E}\nabla_{\boldsymbol{x}}f_{\boldsymbol{\theta}}(\boldsymbol{x})\nabla_{\boldsymbol{x}}^T f_{\boldsymbol{\theta}}(\boldsymbol{x})$, and the expected right singular vectors of the
138 mixed second derivative $\nabla_{\boldsymbol{\theta},\boldsymbol{x}}^2 f_{\boldsymbol{\theta}}(\boldsymbol{x})$. In the case of the text $\omega_n(t) = \alpha t + \beta$. Hence, $n = 1$ and the
139 bound only depends on the gradient covariance and second derivative.

## C  Analytic NAD examples

### C.1  Proofs for linear model of pooling

We first prove the expressions for the example in the text.

**Example 1.** *Let $\phi \sim \mathcal{N}(\mathbf{0}, \sigma_\phi^2 \mathbf{I}_D)$ and $\boldsymbol{\theta} \sim \mathcal{N}(\mathbf{0}, \sigma_{\boldsymbol{\theta}}^2 \mathbf{I}_M)$, the covariance of the input gradient of the linear model of pooling is*

$$\mathbb{E}\nabla_{\boldsymbol{x}} f_{\boldsymbol{\theta},\phi}(\boldsymbol{x}) \nabla_{\boldsymbol{x}}^T f_{\boldsymbol{\theta},\phi}(\boldsymbol{x}) = \sigma_\phi^2 \sigma_{\boldsymbol{\theta}}^2 \operatorname{diag}\left(\boldsymbol{m}^2\right),$$

*and its eigenvectors are the canonical basis elements of $\mathbb{R}^D$, sorted by the entries of $\boldsymbol{m}^2$. Surprisingly, the expected right singular vectors of its mixed second derivative coincide with these eigenvectors,*

$$\mathbb{E}\nabla_{(\boldsymbol{\theta},\phi),\boldsymbol{x}}^2 f(\boldsymbol{x})^T \nabla_{(\boldsymbol{\theta},\phi),\boldsymbol{x}}^2 f(\boldsymbol{x}) = \left(\sigma_{\boldsymbol{\theta}}^2 + \frac{\sigma_\phi^2}{S}\right) \operatorname{diag}(\boldsymbol{m}^2).$$

*This result agrees with what was seen in Sec. 2.2 where we found that the NADs of this architecture are also ranked by $\boldsymbol{m}^2$.*

*Proof.* Borrowing the gradient computations from Sec. B.2,

$$\begin{aligned}
\mathbb{E}_{\boldsymbol{\theta}} \nabla_{\boldsymbol{x}} f_{\boldsymbol{\theta}}(\boldsymbol{x}) \nabla_{\boldsymbol{x}} f_{\boldsymbol{\theta}}(\boldsymbol{x})^T &= \mathbb{E}_{\boldsymbol{\theta},\phi}[(\boldsymbol{A}^T \boldsymbol{\theta}) \odot (\phi \odot \boldsymbol{m})][(\phi^T \odot \boldsymbol{m}^T) \odot (\boldsymbol{\theta}^T \boldsymbol{A})] \\
&= \mathbb{E}_\phi \phi \phi^T \odot \mathbb{E}_{\boldsymbol{\theta}} \boldsymbol{A}^T \boldsymbol{\theta} \boldsymbol{\theta}^T \boldsymbol{A} \odot \boldsymbol{m} \boldsymbol{m}^T = \sigma_\phi^2 \sigma_{\boldsymbol{\theta}}^2 \left(\boldsymbol{I} \odot \boldsymbol{A}^T \boldsymbol{A} \odot \boldsymbol{m} \boldsymbol{m}^T\right) \\
&= \sigma_\phi^2 \sigma_{\boldsymbol{\theta}}^2 \operatorname{diag}\left(\boldsymbol{m}^2\right).
\end{aligned}$$

Similarly, the mixed second derivatives for this model are

$$\nabla_{\boldsymbol{\theta},\boldsymbol{x}}^2 f(\boldsymbol{x}) = \operatorname{diag}(\phi \odot \boldsymbol{m}) \boldsymbol{A}^T$$
$$\nabla_{\phi,\boldsymbol{x}}^2 f(\boldsymbol{x}) = \operatorname{diag}\left((\boldsymbol{A}^T \boldsymbol{\theta}) \odot \boldsymbol{m}\right)$$

which can be combined in

$$\nabla_{(\boldsymbol{\theta},\phi),\boldsymbol{x}}^2 f(\boldsymbol{x}) = \begin{bmatrix} \operatorname{diag}\left((\boldsymbol{A}^T \boldsymbol{\theta}) \odot \boldsymbol{m}\right) \\ \boldsymbol{A} \operatorname{diag}(\phi \odot \boldsymbol{m}) \end{bmatrix}.$$

We can extract its right singular vectors from the eigendecomposition of

$$\begin{aligned}
\mathbb{E}\nabla_{(\boldsymbol{\theta},\phi),\boldsymbol{x}}^2 f(\boldsymbol{x})^T \nabla_{(\boldsymbol{\theta},\phi),\boldsymbol{x}}^2 f(\boldsymbol{x}) &= \mathbb{E}\left[\operatorname{diag}\left((\boldsymbol{A}^T \boldsymbol{\theta})^2 \odot \boldsymbol{m}^2\right)\right] \\
&\quad + \mathbb{E}\left[\operatorname{diag}\left(\phi \odot \boldsymbol{m}\right) \boldsymbol{A}^T \boldsymbol{A} \operatorname{diag}\left(\phi \odot \boldsymbol{m}\right)\right] = \\
&= \sigma_{\boldsymbol{\theta}}^2 \operatorname{diag}(\boldsymbol{m}^2) + \frac{\sigma_\phi^2}{S} \operatorname{diag}(\boldsymbol{m}^2) \\
&= \left(\sigma_{\boldsymbol{\theta}}^2 + \frac{\sigma_\phi^2}{S}\right) \operatorname{diag}(\boldsymbol{m}^2).
\end{aligned}$$

$\square$

### C.2  More examples

We provide a few more examples showing that the gradient covariance can indeed capture the NADs of an architecture.

**Example 2** (Logistic regression). *Let $f_{\boldsymbol{\theta}}(\boldsymbol{x}) = \boldsymbol{\theta}^T \boldsymbol{x}$ be a single layer neural network, i.e., logistic regression. The gradient covariance of this architecture is*

$$\mathbb{E}\nabla_{\boldsymbol{x}} f_{\boldsymbol{\theta}}(\boldsymbol{x}) \nabla_{\boldsymbol{x}}^T f_{\boldsymbol{\theta}}(\boldsymbol{x}) = \sigma_{\boldsymbol{\theta}}^2 \boldsymbol{I}_D.$$

*Because the eigendecomposition of $\boldsymbol{I}_D$ is isotropic, we can see that the logistic regression has no directional bias.*

161 **Example 3** (Single hidden-layer neural network). *Let $f_{\boldsymbol{\theta},\boldsymbol{\Phi}}(\boldsymbol{x}) = \boldsymbol{\theta}^T \rho\left(\boldsymbol{\Phi}^T \boldsymbol{x}\right)$ be a single hidden*
162 *layer neural network with no bias and a ReLU non-linearity $\rho(\cdot)$. Its gradient covariance is*

$$\mathbb{E}\nabla_{\boldsymbol{x}} f_{\boldsymbol{\theta},\boldsymbol{\Phi}}(\boldsymbol{x})\nabla_{\boldsymbol{x}}^T f_{\boldsymbol{\theta},\boldsymbol{\Phi}}(\boldsymbol{x}) = \frac{\sigma_{\boldsymbol{\theta}}^2 \sigma_{\boldsymbol{\Phi}}^2}{2},$$

163 *and we see that this architecture has also no directional bias.*

164 *Proof.* The gradient of $f_{\boldsymbol{\theta},\boldsymbol{\Phi}}(\boldsymbol{x})$ is $\nabla_{\boldsymbol{x}} f_{\boldsymbol{\theta},\boldsymbol{\Phi}}(\boldsymbol{x}) = \boldsymbol{\Phi}\,\mathrm{diag}\left(\rho'\left(\boldsymbol{\Phi}^T \boldsymbol{x}\right)\right)\boldsymbol{\theta}$, where the derivative of the
165 ReLU non-linearity is the indicator function $\rho'(\boldsymbol{u}) = \mathbb{1}_{\boldsymbol{u}\succeq\boldsymbol{0}}$.

166 Hence,

$$\mathbb{E}\nabla_{\boldsymbol{x}} f_{\boldsymbol{\theta},\boldsymbol{\Phi}}(\boldsymbol{x})\nabla_{\boldsymbol{x}}^T f_{\boldsymbol{\theta},\boldsymbol{\Phi}}(\boldsymbol{x}) = \mathbb{E}\left[\boldsymbol{\Phi}\,\mathrm{diag}\left(\rho'\left(\boldsymbol{\Phi}^T \boldsymbol{x}\right)\right)\boldsymbol{\theta}\boldsymbol{\theta}^T\,\mathrm{diag}\left(\rho'\left(\boldsymbol{\Phi}^T \boldsymbol{x}\right)\right)\boldsymbol{\Phi}^T\right] =$$
$$= \sigma_{\boldsymbol{\theta}}^2 \mathbb{E}\left[\boldsymbol{\Phi}\,\mathrm{diag}\left(\rho'\left(\boldsymbol{\Phi}^T \boldsymbol{x}\right)\right)\mathrm{diag}\left(\rho'\left(\boldsymbol{\Phi}^T \boldsymbol{x}\right)\right)\boldsymbol{\Phi}^T\right] =$$
$$= \sigma_{\boldsymbol{\theta}}^2 \mathbb{E}\left[\boldsymbol{\Phi}\,\mathrm{diag}\left(\mathbb{1}_{\boldsymbol{\Phi}^T \boldsymbol{x}\succeq\boldsymbol{0}}\right)\boldsymbol{\Phi}^T\right]$$

167 This expectation can be computed analytically. In particular note that

$$\mathbb{E}\left[\boldsymbol{\Phi}\,\mathrm{diag}\left(\mathbb{1}_{\boldsymbol{\Phi}^T \boldsymbol{x}\succeq\boldsymbol{0}}\right)\boldsymbol{\Phi}^T\right][i,j] = \sum_{k=1}^{D}\mathbb{E}\left[\boldsymbol{\Phi}[i,k]\boldsymbol{\Phi}[j,k]\mathbb{1}_{\boldsymbol{\Phi}[i,:]^T \boldsymbol{x}\geq 0}\right].$$

168 Therefore, if $i \neq j$

$$\mathbb{E}\left[\boldsymbol{\Phi}[i,k]\boldsymbol{\Phi}[j,k]\mathbb{1}_{\boldsymbol{\Phi}[i,:]^T \boldsymbol{x}\geq 0}\right] = \mathbb{E}_{\boldsymbol{\Phi}[i,k]}\left[\mathbb{E}_{\boldsymbol{\Phi}[j,k]}\left[\boldsymbol{\Phi}[i,k]\boldsymbol{\Phi}[j,k]\mathbb{1}_{\boldsymbol{\Phi}[i,:]^T \boldsymbol{x}\geq 0}\middle|\boldsymbol{\Phi}[i,k]\right]\right] = 0.$$

169 On the other hand, when $i = j$,

$$\mathbb{E}\left[\boldsymbol{\Phi}\,\mathrm{diag}\left(\mathbb{1}_{\boldsymbol{\Phi}^T \boldsymbol{x}\succeq\boldsymbol{0}}\right)\boldsymbol{\Phi}^T\right][i,i] = \sum_{k=1}^{D}\mathbb{E}\left[\boldsymbol{\Phi}^2[i,k]\mathbb{1}_{\boldsymbol{\Phi}[i,:]^T \boldsymbol{x}\geq 0}\right]$$
$$= \mathbb{E}\left[\|\boldsymbol{\Phi}[i,:]\|^2\mathbb{1}_{\boldsymbol{\Phi}[i,:]^T \boldsymbol{x}\geq 0}\right].$$

170 Let $p(\boldsymbol{w})$ denote the probability density function of a Gaussian random vector $\boldsymbol{w} \sim \mathcal{N}(\boldsymbol{0}, \sigma^2 \boldsymbol{I})$ and
171 $\boldsymbol{U} \in \mathrm{SO}(D)$ and orthonormal matrix such that $\boldsymbol{x}' = \boldsymbol{U}^T \boldsymbol{x}$ with $\boldsymbol{x}'[1] = \|\boldsymbol{x}\|$ and $\boldsymbol{x}'[i] = 0$ for
172 $i = 2, \ldots, D$. Then,

$$\langle \boldsymbol{w}, \boldsymbol{x}\rangle \geq 0 \Leftrightarrow \langle \boldsymbol{U}\boldsymbol{w}, \boldsymbol{x}\rangle \geq 0\rangle \Leftrightarrow \langle \boldsymbol{w}, \boldsymbol{U}^T \boldsymbol{x}\rangle \geq 0 \Leftrightarrow \boldsymbol{w}[1]\|\boldsymbol{x}\|_2 \geq 0 \Leftrightarrow \boldsymbol{w}[1] \geq 0.$$

173 Using this equivalence, we can compute the expectation

$$\mathbb{E}\left[\|\boldsymbol{w}\|^2\mathbb{1}_{\boldsymbol{w}^T \boldsymbol{x}\geq 0}\right] = \int_{\mathbb{R}^D}\mathbb{1}_{\boldsymbol{w}^T \boldsymbol{x}\geq 0}\|\boldsymbol{w}\|^2 p(\boldsymbol{w})d\boldsymbol{w} = \int_{\mathbb{R}^D}\mathbb{1}_{\boldsymbol{w}[1]\geq 0}\|\boldsymbol{w}\|^2 p(\boldsymbol{w})d\boldsymbol{w} =$$
$$= \int_{\mathbb{R}^D}\mathbb{1}_{\boldsymbol{w}[1]\geq 0}\boldsymbol{w}^2[1]p(\boldsymbol{w})d\boldsymbol{w} + \sum_{i=2}^{D}\int_{\mathbb{R}^D}\mathbb{1}_{\boldsymbol{w}[1]\geq 0}\boldsymbol{w}^2[i]p(\boldsymbol{w})d\boldsymbol{w} =$$
$$= \int_{0}^{+\infty}\boldsymbol{w}^2[1]\frac{1}{\sqrt{2\pi\sigma^2}}e^{-\frac{\boldsymbol{w}^2[1]}{2\sigma^2}}d\boldsymbol{w}[1] +$$
$$+ \frac{D-1}{2}\int_{-\infty}^{+\infty}\boldsymbol{w}^2[2]\frac{1}{\sqrt{2\pi\sigma^2}}e^{-\frac{\boldsymbol{w}^2[2]}{2\sigma^2}}d\boldsymbol{w}[2] =$$
$$= \frac{1}{2}\sigma^2 + \frac{D-1}{2}\sigma^2 = \frac{D}{2}\sigma^2.$$

174 Plugging this into the expressions of the gradient covariance we get

$$\mathbb{E}\nabla_{\boldsymbol{x}} f_{\boldsymbol{\theta},\boldsymbol{\Phi}}(\boldsymbol{x})\nabla_{\boldsymbol{x}}^T f_{\boldsymbol{\theta},\boldsymbol{\Phi}}(\boldsymbol{x}) = \sigma_{\boldsymbol{\theta}}^2 \mathbb{E}\left[\boldsymbol{\Phi}\,\mathrm{diag}\left(\mathbb{1}_{\boldsymbol{\Phi}^T \boldsymbol{x}\succeq\boldsymbol{0}}\right)\boldsymbol{\Phi}^T\right] =$$
$$= \sigma_{\boldsymbol{\theta}}^2 \mathbb{E}\left[\|\boldsymbol{\Phi}[i,:]\|^2\mathbb{1}_{\boldsymbol{\Phi}[i,:]^T \boldsymbol{x}\geq 0}\right]\boldsymbol{I}_D =$$
$$= \frac{D}{2}\sigma_{\boldsymbol{\theta}}^2 \sigma_{\boldsymbol{\Phi}}^2 \boldsymbol{I}_D.$$

175 $\square$

176 **Example 4** (Non-linear model of pooling). *Let $f_{\boldsymbol{\theta},\boldsymbol{\phi}}(\boldsymbol{x}) = \boldsymbol{\theta}^T \boldsymbol{A}(\boldsymbol{m} \odot \rho(\boldsymbol{\phi} \odot \boldsymbol{v}))$ with $\nabla_{\boldsymbol{x}} f_{\boldsymbol{\theta},\boldsymbol{\phi}}(\boldsymbol{x}) =$*
177 *$(\boldsymbol{A}^T \boldsymbol{\theta}) \odot (\rho'(\boldsymbol{\phi} \odot \boldsymbol{x}) \odot \boldsymbol{\phi} \odot \boldsymbol{m})$. Then,*

$$\mathbb{E}\nabla_{\boldsymbol{x}} f_{\boldsymbol{\theta},\boldsymbol{\phi}}(\boldsymbol{x})\nabla_{\boldsymbol{x}}^T f_{\boldsymbol{\theta},\boldsymbol{\phi}}(\boldsymbol{x}) = \sigma_{\boldsymbol{\phi}}^2 \sigma_{\boldsymbol{\theta}}^2 \left(\boldsymbol{A}^T \boldsymbol{A} \odot \boldsymbol{m}\boldsymbol{m}^T \odot \boldsymbol{\Xi}(\boldsymbol{x})\right),$$

178 *where $\boldsymbol{\Xi}(\boldsymbol{x}) \in \mathbb{R}^{D \times D}$ is a matrix that depends on the input vector $\boldsymbol{x}$ and can be computed in closed*
179 *form.*

180 *In particular, if the distribution of $\boldsymbol{x}$ is symmetric around $\boldsymbol{0}$, then $\mathbb{E}\,\boldsymbol{\Xi}(\boldsymbol{x}) = \boldsymbol{I}_D$ and the average*
181 *gradient covariance with respect to the input would be identical to that of the linear model of pooling.*

182 *Proof.* Expanding the covariance definition

$$\mathbb{E}\nabla_{\boldsymbol{x}} f_{\boldsymbol{\theta},\boldsymbol{\phi}}(\boldsymbol{x})\nabla_{\boldsymbol{x}}^T f_{\boldsymbol{\theta},\boldsymbol{\phi}}(\boldsymbol{x}) = \mathbb{E}[(\boldsymbol{A}^T \boldsymbol{\theta}) \odot (\rho'(\boldsymbol{\phi} \odot \boldsymbol{x}) \odot \boldsymbol{\phi} \odot \boldsymbol{m})]$$
$$[(\rho'(\boldsymbol{\phi}^T \odot \boldsymbol{x}^T) \odot \boldsymbol{\phi}^T \odot \boldsymbol{m}^T) \odot (\boldsymbol{\theta}^T \boldsymbol{A})] =$$
$$= \mathbb{E}[(\rho'(\boldsymbol{\phi} \odot \boldsymbol{x}) \odot \boldsymbol{\phi}][(\rho'(\boldsymbol{\phi}^T \odot \boldsymbol{x}^T) \odot \boldsymbol{\phi}^T] \odot \mathbb{E}\boldsymbol{A}^T \boldsymbol{\theta}\boldsymbol{\theta}^T \boldsymbol{A} \odot \boldsymbol{m}\boldsymbol{m}^T.$$

183 We can see that the only difference with respect to the linear model case is the first expectation. Let
184 $\boldsymbol{\Xi}(\boldsymbol{x}) \in \mathbb{R}^{D \times D}$ be the matrix with entries

$$\boldsymbol{\Xi}[i,j] = \mathbb{E}[(\rho'(\boldsymbol{\phi} \odot \boldsymbol{x}) \odot \boldsymbol{\phi}][(\rho'(\boldsymbol{\phi}^T \odot \boldsymbol{x}^T) \odot \boldsymbol{\phi}^T][i,j]$$

$$= \mathbb{E}[\boldsymbol{\phi}[i]\boldsymbol{\phi}[j]\mathbb{1}_{\boldsymbol{\phi}[i]\boldsymbol{x}[i]\geq 0}\mathbb{1}_{\boldsymbol{\phi}[j]\boldsymbol{x}[]j]\geq 0}] = \begin{cases} \mathbb{E}[\boldsymbol{\phi}[i]\mathbb{1}_{\boldsymbol{\phi}[i]\boldsymbol{x}[i]\geq 0}]\mathbb{E}[\boldsymbol{\phi}[j]\mathbb{1}_{\boldsymbol{\phi}[j]\boldsymbol{x}[j]\geq 0}] & i \neq j \\ \mathbb{E}[\boldsymbol{\phi}^2[j]\mathbb{1}_{\boldsymbol{\phi}[j]\boldsymbol{x}[j]\geq 0}] & i = j \end{cases}$$

185 Depending on $\boldsymbol{x}$ the expectation $\mathbb{E}[\boldsymbol{\phi}[j]\mathbb{1}_{\boldsymbol{\phi}[j]\boldsymbol{x}[j]\geq 0}]$ takes different values:

$$\mathbb{E}[\boldsymbol{\phi}[j]\mathbb{1}_{\boldsymbol{\phi}[j]\boldsymbol{x}[j]\geq 0}] = \begin{cases} \dfrac{\sigma_{\boldsymbol{\phi}}\sqrt{2}}{2\sqrt{\pi}} & \boldsymbol{x}[j] > 0 \\ -\dfrac{\sigma_{\boldsymbol{\phi}}\sqrt{2}}{2\sqrt{\pi}} & \boldsymbol{x}[j] < 0 \\ 0 & \boldsymbol{x}[i] = 0 \end{cases}$$

186 Similarly

$$\mathbb{E}[\boldsymbol{\phi}^2[j]\mathbb{1}_{\boldsymbol{\phi}[j]\boldsymbol{x}[j]\geq 0}] = \begin{cases} \dfrac{\sigma_{\boldsymbol{\phi}}^2}{2}\left(1 - \dfrac{2}{\pi}\right) & \boldsymbol{x}[j] \neq 0 \\ \sigma_{\boldsymbol{\phi}}^2 & \boldsymbol{x}[j] = 0 \end{cases}$$

187 Then the covariance depending on $\boldsymbol{x}$ becomes,

$$\mathbb{E}\nabla_{\boldsymbol{x}} f_{\boldsymbol{\theta},\boldsymbol{\phi}}(\boldsymbol{x})\nabla_{\boldsymbol{x}}^T f_{\boldsymbol{\theta},\boldsymbol{\phi}}(\boldsymbol{x}) = \sigma_{\boldsymbol{\phi}}^2 \sigma_{\boldsymbol{\theta}}^2 \left(\boldsymbol{A}^T \boldsymbol{A} \odot \boldsymbol{m}\boldsymbol{m}^T \odot \boldsymbol{\Xi}(\boldsymbol{x})\right).$$

188 $\qquad\qquad\qquad\qquad\qquad\qquad\qquad\qquad\qquad\qquad\qquad\qquad\qquad\qquad\qquad\qquad\qquad\quad$ $\square$

## D  NADs of CNNs

As highlighted in Sec. 3.1, we can use two algorithms to identify the NADs of an architecture without training. Surprisingly, both algorithms yield very similar results, but the algorithm based on the eigendecomposition of the gradient covariance is numerically much more stable. Indeed, for most randomly initialized networks, the norm of the second derivative with respect to the weights and input is very small, rendering the numerical singular value decomposition of the second derivative very unstable. Meanwhile, the gradient covariance only requires information about first order gradients and these are orders of magnitudes larger than the second derivatives. For this reason, in all our experiments we used the eigenvectors of the gradient covariance as approximations of the NADs of a given architecture.

We provide now the implementation details of both algorithms, as well as some examples of NADs identified with both methods.

### D.1  NADs obtained through the eigendecomposition of the gradient covariance

Algorithm 1 describes the steps required to identify the NADs of an architecture using its input gradient covariance. As we can see, this procedure amounts to sampling $T$ architectures from its weight initialization distribution, computing its input gradient at an arbitrary input point $\boldsymbol{x}$, and performing a Principal Component Analysis on the gradient samples.

---

**Algorithm 1** NAD discovery through gradient covariance

---

**Require:** Network architecture $f_{\boldsymbol{\theta}}$, parameter distribution $\boldsymbol{\Theta}$, evaluation sample $\boldsymbol{x}$, number of Monte-Carlo samples $T$, and finite-difference scale $h$.
1: $\mathcal{G} \leftarrow \varnothing$                                                                  ▷ Gradient samples
2: **for** $t = 1, \dots, T$ **do**
3:     Draw $\boldsymbol{\theta} \sim \boldsymbol{\Theta}$
4:     $\bar{\nabla}_{\boldsymbol{x}} f_{\boldsymbol{\theta}}(\boldsymbol{x}) \leftarrow \mathbf{0}$
5:     **for** $i = 1, \dots, D$ **do**
6:         $\bar{\nabla}_{\boldsymbol{x}} f_{\boldsymbol{\theta}}(\boldsymbol{x})[i] \leftarrow \dfrac{f_{\boldsymbol{\theta}}(\boldsymbol{x} + h\boldsymbol{e}_i) - f_{\boldsymbol{\theta}}(\boldsymbol{x} - h\boldsymbol{e}_i)}{2h}$    ▷ Compute finite difference gradient
7:     **end for**
8:     $\mathcal{G} \leftarrow \mathcal{G} \cup \bar{\nabla}_{\boldsymbol{x}} f_{\boldsymbol{\theta}}(\boldsymbol{x})$
9: **end for**
10: $\{(\boldsymbol{u}_i, \lambda_i)\}_{i=1}^{D} \leftarrow \text{PCA}(\mathcal{G})$                         ▷ Perform Principal Component Analysis
11: **return** $\{\boldsymbol{u}_i\}_{i=1}^{D}$

---

In practice, we found out that using finite differences with a scale of $h = 100$ to approximate the gradients instead of backpropagation was necessary to obtain meaningful results. We believe the reason for this is that the finite differences allow to capture a coarser scale of the function geometry and hide the effect of higher order terms, as they do not rely on very local fluctuations of the input geometry. We leave for future research the understanding of this phenomenon.

We now show some additional examples of NADs obtained using Algorithm 1 on a LeNet, VGG-11, ResNet-18 and DenseNet121.

 **D.1.1    LeNet**

Figure S5: NADs of LeNet obtained through eigendecomposition of gradient covariance

 **D.1.2    VGG11**

Figure S6: NADs of VGG16 obtained through eigendecomposition of gradient covariance

 **D.1.3    ResNet-18**

Figure S7: NADs of ResNet-18 obtained through eigendecomposition of gradient covariance

 **D.1.4 DenseNet-121**

Figure S8: NADs of DenseNet-121 obtained through eigendecomposition of gradient covariance

## D.2 NADs obtained through the SVD of the mixed second derivative

The second way we can identify the NADs without training is using the expected right singular vectors of the mixed second derivative, $\nabla^2_{\boldsymbol{x},\boldsymbol{\theta}} f_{\boldsymbol{\theta}}(\boldsymbol{x})$. However, note that the mixed second derivative has a number of entries equal to the product of the weight and input dimensionalities, which can amount to more than a trillion elements. This makes it impossible to store this object in any common computational platform, and hence we can only estimate its singular vectors using power iteration methods [2]. Specifically, these methods estimate the spectral decomposition of a linear operator by sequentially alternating between the application of the linear operator on a vector and its adjoint.

Consequently, we just need an efficient way to compute $\nabla^2_{\boldsymbol{x},\boldsymbol{\theta}} f_{\boldsymbol{\theta}(\boldsymbol{x})} \boldsymbol{v}$ and $\boldsymbol{v}'^{T} \nabla^2_{\boldsymbol{x},\boldsymbol{\theta}} f_{\boldsymbol{\theta}(\boldsymbol{x})}$ for any $\boldsymbol{v}$ and $\boldsymbol{v}'$ to be able to compute the SVD. Algorithm 2 details these procedures. As we can see, in our algorithms we use a finite difference approximation to compute the directional input derivative of $\nabla_{\boldsymbol{\theta}} f_{\boldsymbol{\theta}}(\boldsymbol{x})$. Again, this helps for stability of the results.

---

**Algorithm 2** NAD discovery through mixed second derivative

---

**Require:** Network architecture $f_{\boldsymbol{\theta}}$, parameter distribution $\boldsymbol{\Theta}$, evaluation sample $\boldsymbol{x}$, number of Monte-Carlo samples $T$, and finite-difference scale $h$.

1: **procedure** DVP($\mathcal{F}$, $\boldsymbol{v}$)$\qquad\qquad\qquad\qquad\qquad\qquad$ ▷ Computes $\nabla^2_{\boldsymbol{x},\boldsymbol{\theta}} f_{\boldsymbol{\theta}(\boldsymbol{x})} \boldsymbol{v}$
2: $\qquad$ **for** $f_{\boldsymbol{\theta}} \in \mathcal{F}$ **do**
3: $\qquad\qquad d \leftarrow 0$
4: $\qquad\qquad d \leftarrow d + \dfrac{\nabla_{\boldsymbol{\theta}} f_{\boldsymbol{\theta}}(\boldsymbol{x} + h\boldsymbol{v}) - \nabla_{\boldsymbol{\theta}} f_{\boldsymbol{\theta}}(\boldsymbol{x} - h\boldsymbol{v})}{2h}$
5: $\qquad$ **end for**
6: $\qquad$ **return** $d/T$
7: **end procedure**

8: **procedure** ADVP($\mathcal{F}$, $\boldsymbol{v}'$)$\qquad\qquad\qquad\qquad\qquad\qquad$ ▷ Computes $\boldsymbol{v}'^{T} \nabla^2_{\boldsymbol{x},\boldsymbol{\theta}} f_{\boldsymbol{\theta}(\boldsymbol{x})}$
9: $\qquad$ **for** $f_{\boldsymbol{\theta}} \in \mathcal{F}$ **do**
10: $\qquad\qquad d \leftarrow 0$
11: $\qquad\qquad d \leftarrow d + \nabla_{\boldsymbol{x}} \left( \boldsymbol{v}'^{T} \nabla_{\boldsymbol{\theta}} f_{\boldsymbol{\theta}}(\boldsymbol{x}) \right)$
12: $\qquad$ **end for**
13: $\qquad$ **return** $d/T$
14: **end procedure**

15: $\mathcal{F} \leftarrow \varnothing$ $\qquad\qquad\qquad\qquad\qquad\qquad\qquad\qquad\qquad\qquad\qquad$ ▷ Function samples
16: **for** $t = 1, \ldots, T$ **do**
17: $\qquad$ Draw $\boldsymbol{\theta} \sim \boldsymbol{\Theta}$
18: $\qquad$ $\mathcal{F} \leftarrow \mathcal{F} \cup f_{\boldsymbol{\theta}}$
19: **end for**

20: $\{(\boldsymbol{u}_i, \sigma_i)\} \leftarrow$ PowerIteration(DVP, ADVP)$\qquad$ ▷ SVD through power iterations
21: **return** $\{\boldsymbol{u}_i\}_{i=1}^{D}$

---

In the next figures, we show the results of the application of these algorithm to a LeNet, VGG-10 and ResNet-18. However, due to the high computational complexity of Algorithm 2 on large networks, we do not show them for the larger DenseNet-121. At this stage, it is important to highlight that the results of Algorithm 2 are much noisier than those of Algorithm 1 (as seen in the resulting NADs depicted in Sec. D.1 and Sec. D.2). We believe this is due to the bad conditioning of Algorithm 2 due to the small magnitude of the second derivatives and the use of a power iteration method in Algorithm 2 with respect to the exact eigendecomposition in Algorithm 1. Nevertheless, looking at the shape (especially in the spectral domain) of the first few NADs obtained with both algorithms we can see that they are indeed very aligned.

 **D.2.1   LeNet**

Figure S9: NADs of LeNet obtained through SVD of mixed second derivative

39 **D.2.2 VGG11**

Figure S10: NADs of VGG16 obtained through SVD of mixed second derivative

 **D.2.3    ResNet-18**

Figure S11: NADs of ResNet-18 obtained through SVD of mixed second derivative

241    **D.3    Further experiments with NADs**

242    We now provide some further experiments using the NADs of some common neural network archi-
243    tectures. First, we give two additional experiments on the performance of a VGG11, and a multilayer
244    perceptron (MLP) with 3 hidden layers with 500 neurons each, on a sequence of linearly separable
245    datasets aligned with its NADs. As we can see in Fig S12, the VGG11 behaves very similarly to the
246    other CNNs (see Fig. 6), only being able to generalize to a few distributions, whereas the MLP can
247    always perfectly generalize to the test distribution. Note also, that the eigenvalue decay on the MLP is
248    much less pronounced. In fact, we believe that this is only a result of the finite set of gradient samples
249    used to perform its eigendecomposition, and we conjecture that in the limit of infinite samples the
250    eigenvalue distribution of the MLP will be completely flat (as we formally proved for the single
251    hidden layer network of Example 3).

252    **D.3.1    Speed of convergence**

253    NADs also have an effect in optimization. To show this, we tracked the training loss of a LeNet and
254    a ResNet-18 when trained on different $\mathcal{D}(v)$ parameterized by the NAD sequence. Fig. S13 shows
255    these results. As expected, even if in all cases these networks achieved almost a $100\%$ test accuracy,

Figure S12: **(Green)** Normalized covariance eigenvalues and **(brown)** test accuracies of a MLP and a VGG11 trained on linearly separable distributions parameterized by their NADs. ($\sigma = 3$, $\epsilon = 1$)

the effect of NADs is clearly visible during optimization. This is, it takes much longer for these networks to converge to small training losses when the discriminative information of the dataset is aligned with the later NADs as opposed to the first ones. This is similar to the phenomenon described in Fig. 4b where we identified the same behaviour with respect to the Fourier basis. However, in that case, higher frequency was not a direct indicator of training hardness (cf. NAD index).

(a) LeNet ($\sigma = 0$)

(b) ResNet-18 ($\sigma = 1$)

Figure S13: Training loss per batch of different networks trained using different training sets drawn from $\mathcal{D}(\boldsymbol{v})$ ($\epsilon = 1$, and $\sigma$ chosen to accentuate differences). Directions $\boldsymbol{v}$ taken from the NAD sequence.

### D.3.2 Generalization vs. number of training samples

NADs encapsulate the preference of a network to search for discriminative features in some particular directions. This means that a network first tries to fit the training data using features aligned with NADs of lower indices, before proceeding to later ones. In that sense, and for a fixed level of noise $\sigma$, one can argue that, if the discriminative direction of the data is aligned with a NAD of higher index (i.e., not properly aligned with the directional inductive bias of the network), it is quite likely that the network will overfit to some discriminative but non-generalizing solutions, using noisy features that are aligned with NADs of lower indices. In this case, and for reducing such spurious correlations, more training samples might be necessary for the network to "ignore" such solutions

and seek for other discriminative ones using NADs of higher indices (and hence eventually finding the discriminative and generalizing one).

On the contrary, if the discriminative direction of the data is aligned with a lower NAD index (i.e., properly aligned with the directional inductive bias of the network), then the network tries to fit the training data along the truly generalizing direction earlier; hence, the possibility of overfitting to noisy features appearing along higher NAD indices is reduced. In that sense, even a few training samples might be enough for the network to converge to the generalizing solution.

An illustration of this dependency between the alignment of the generalizing direction with the NADs, and the number of training samples, is shown in Fig. S14. For both cases, it is clear that less training data are required for the network to generalize when the discriminative direction $v$ is aligned with the lower NADs of the network. On the contrary, when $v$ is aligned with higher NADs, more data is required for the network to "ignore" the noisy features and find the generalizing solution. In fact, as clearly observed for the case of ResNet-18, given a large amount of training samples (considering the simplicity of the task) the network can eventually generalize perfectly, regardless the position of the direction $v$.

(a) LeNet

(b) ResNet-18

Figure S14: Generalization vs number of training samples for two CNNs trained using different training sets drawn from $\mathcal{D}(v)$ ($\epsilon = 1$, and $\sigma = 3$). Directions $v$ taken from the NAD sequence.

## E  Details of experiments on CIFAR10

All our experiments on CIFAR-10 use networks trained for $50$ epochs using SGD with a linearly decaying learning rate with maximum value $0.21$, fixed momentum $0.9$ and a weight decay of $5 \times 10^{-4}$. Again, our objective is not to obtain the best achievable performance, but to show relative differences with respect to NADs for a fixed training setup. Hence, the hyperparameters of these networks were not optimized in any way during this work. Instead they were selected from a set of best practices from the DAWNBench submissions that have been empirically shown to give a good trade-off in terms of convergence speed and performance.

We finish this section with a detailed description of the poisoning experiment. In particular, recall that, in the binary class setting, i.e., $y \in \{-1, +1\}$ an easy way to introduce a poisonous carrier on a sample $x$ is to substitute the information on that sample in a given direction by $\epsilon y$. However, this means that, for a given direction $u$, we can only allocate at most two classes. A simple extension to the multi-class case, i.e., $y \in \{1, \ldots, L\}$, uses therefore $\lceil L/2 \rceil$ directions to poison all samples.

CIFAR-10 has $L = 10$ classes, but also, its samples contain information spread along $K = 3$ color channels. The NADs that we computed in Sec. 3.1 were computed for single-channel inputs. Hence, we need to extend them to work in the $K$-channel case. Let $\{u_i\}_{i=1}^{D}$ be the NADs of a deep neural network for a single channel input. The NADs of the same architecture with $K$ input channels are $\{u_i \otimes e'_k, \ i = 1, \ldots, D, \ k = 1, \ldots, K\}$, where $e'_k$ represents the $k^{\text{th}}$ canonical basis vector of $\mathbb{R}^K$.

All in all, using these extensions to the simple setting, we can easily poison CIFAR-10. Given a carrier NAD index $i$, for each sample $x_j \in \mathbb{R}^{DK}$ in the training set with associated label $y_j$ we can modify it such that it satisfies $x_j^T (u_i \otimes e'_{\lfloor y_j/2 \rfloor}) = \epsilon(2[\![y_j]\!]_2 - 1)$. Note that, for any $\epsilon > 0$, this small modification on the training set renders each class linearly separable from the others using only the poisonous features. However a classifier that uses these features will not be able to generalize to the unpoisoned test set.

## Footnotes

[1]For more information about the properties of the 2D-DFT, we refer the reader to [1].