[Reviews · NeurIPS 2020]

Review 1

Summary and Contributions: The paper describes the model-driven inductive bias that controls the generalization in deep neural networks. For a given architecture, it uses neural anisotropy directions (NADs) vectors that encapsulate the directional inductive bias of that architecture. These NADs are specific to each architecture and thus act as a signature and encode the preference of a given network to separate the input data based on some particular features. To validate this, the framework considers the classification of linearly separable distributions and proves that many CNNs have hard time in solving this simple task based on the direction of the discriminative feature of the given distributions. The validation of NADs is carried out using LeNet, RseNet18 and DenseNet121 CNN architectures and their directional inductive biases using CIFAR-10 dataset.

Strengths: A novel framework for model-driven directional inductive bias that controls generalization in deep neural networks. This work is relevant to NIPS/AI/Computer Vision communities. It gives insight into a deep models’ decision-making process. The mathematical derivation and the respective experimental evaluation suggest that this bias is encoded by an orthonormal set of vectors for each architecture and is named as neural anisotropy directions (NADs) vectors. These NADs are responsible for the selection of discriminative features used by CNNs to separate a training set. The experimental validation is carried out using three different types of CNN architectures (LeNet, RseNet18 and DenseNet121). Theoretical background of anisotropic loss of information and the anisotropic conditioning of the optimization landscape is provided to support the computation of NADs. Finally, the paper also provides some examples of NADs. The paper experimentally validates the role of NADs on generalization in different scenarios such as learning linearly separable datasets and learning CIFR-10 dataset.

Weaknesses: The theoretical description of directional inductive is very good but one would like to see more experimental validation of this approach using another dataset (other than CIFR-10). I can follow the problem and the theory behind the directional inductive bias. However, I got lost in the section 2. This is mainly I am not able to establish the link between the anisotropic loss of information and the anisotropic conditioning of the optimization landscape, and the computation of NADs. In the paper, it is mentioned that the directional bias can be identified based on the speed of the convergence during training. It is unclear to me how and is not explained in the paper. It is also unclear what is NAD indexes and how 1024 NAD index values are identified. Moreover, in learning CIFR-10 dataset, why only ith and (i+1)th indexes to introduce poisonous carrier.

Correctness: The proposed methodology seems to be correct. My main area of research is not on inductive bias. Therefore, I would like to refer the comments from other reviewers.

Clarity: The paper is well-written and easy to follow. However, it could have been improved with a bit more details about establishing relationships between NADs computation and theory behind the directional inductive bias.

Relation to Prior Work: The paper discusses about the important related work. A brief explanation of the key message in the related work is provided. However, there is no relationships/explanation on how the proposed approach differs from these.

Reproducibility: No

Additional Feedback: It is a good piece of work to get insight into the deep models. However, there is lack of clarity in many areas (please refer to the “Weakness” section), as well as the experimental validation is carried out using only one dataset (CIFAR-10). This has significantly impacted the overall rating of this work. I have given major comments in the “Weakness” section and please refer to it. *********** Post-Rebuttal Update ************* I would like to thank authors for their response. I have gone through the rebuttal and it addresses most of my concerns. I have also gone through the reviews of other reviewers. By considering all these information, I am convinced the overall work is good for NIPS2020. Therefore, I stick to my original decision (vote for 7).


Review 2

Summary and Contributions: This paper defines a notion of "neural anisotropy directions (NADs)", which are directions of input differences for which a CNN is biased to be able to easily separate, when the input is linearly separable along those directions. Also the paper propose an efficient method to identify NADs for several CNN architectures without having to train the networks.

Strengths: This paper presents several interesting properties of NADs and new discoveries as follows: - CNNs generalize well when the discriminative information of the data is aligned with NADs - Surprisingly, gigantic DenseNet generalizes to only a few of DFT basis directions (Fig. 1) - Pooling layers are a major source for the directional inductive bias. When all pooling layers are removed most of the bias disappears (Fig. 3) - Training converges fast if input distributions are aligned with NADs (Fig.2) - There are anisotropic loss of information and anisotropic conditioning of the optimization landscape Further, the paper propose a computationally efficient method to estimate NADs which requires no training based on clever idea of using simple discriminative task callsed discrminative dipole.

Weaknesses: - Although this paper gives a precious lesson that the architecture should have an inductive bias that aligns with data distribution, the applicability of NADs to Neural Architecture Search seems not trivial since it is the reverse task of all the experiments done in this paper. - It would have been great if it is possible to improve a certain architecture with increased accuracy based on the idea of NADs. - It would be interesting how NADs change depending on the type of layers used in the CNN. For example, how will NADs change depending on whether the layers have skip connections (as in Residual networks) or not. - This paper suggests that the inductive bias of an architecture can be computed before training, but there is no experiment on how NADs change during training.

Correctness: The reviewer didn't find any flaws yet.

Clarity: The paper is very well written with high clarity.

Relation to Prior Work: Yes, it is satisfactory.

Reproducibility: Yes

Additional Feedback: After author feedback: Authors' feedback resolved most of my concerns, so I keep my positive rating.


Review 3

Summary and Contributions: After rebuttal: I thank the authors for writing for writing a detailed rebuttal and clarifying the take-away from the poisoning experiment and providing further details on the flipping experiemnt. However, I the experiments still doesn't convince me that the existence of NADs has anything to do with the current performance of neural networs (or that the current neural networks rely on NADs for their performance). The existence of NADs is an interesting phenomena but it's impact on real taks is unclear to me. Given this, I'm raising my vote to a weak accept 6. =========== The paper introduces a new framework to look at the inductive biases of CNNs. The authors show that CNNs have a hard time solving a linearly separable classification task if the classification boundary is in certain directions of the input space, and similarly, the task is easier to solve when the boundary aligns with certain other directions. The directions of the input space in which the network favors are called Neural Anisotropy Directions (NAD). The paper shows that the existence of NADs are closely linked to pooling layers in the network and that for popular classification datasets like CIFAR10C, the NADs of commonly used network ResNet-18 aligns with the dataset.

Strengths: In my understanding, the idea of NADs is very novel and very relevant to the community. This gives us a new lens to look at and understand CNN architectures better. 1. The idea of NADs is explained well with the motivating example in Section 2. 2. There's a mathematical characterization of the phenomena on linear models (I haven't looked through the proofs) 3. The paper shows that the NADs of CIFAR10 and popular architectures used on CIFAR10 align.

Weaknesses: 1. The proposed method is computationally very expensive to execute, particularly if one has to do it on much larger dimensional input space. 2. Can you visualize how poisoned CIFAR10 images look at? Are the changes perceptually visible? 3. Related to 2, since CIFAR10 is very small, can you demonstrate this phenomenon on larger datasets? I realize that the method is extremely computationally intensive and for imagenet with 255x255 images and 1000 classes, it's going to be very time-consuming. Can you downsample imagenet images, say 128x128, and use restricted imagenet (with only 16( or 10?, I forget) classes) and demonstrate similar conclusions? 4. I agree with the conclusions of Fig 7. If you make the dataset linearly separable along some directions, the network is able to pick it up and if you make it linearly separable among some other directions, the network is not able to pick it up (similar to section 2). However, I disagree that this implies that NADs of the CNNs align with that for CIFAR10 dataset. I don't see why there should be a discriminative feature for CIFAR10 along the direction of NADs. When the last NADs are poisoned, in the absence of easy discriminative features in the training dataset, the CNN learns the real discriminatory feature and hence generalize. When initial NADs are poisoned, the CNNs just latch on to easy NADs and do not generalize. I'm not sure what is the conclusion of this experiment which relates to CIFAR10 - I think you'll observe the same phenomenon even if you replace CIFAR10 with a synthetic mixture of Gaussian dataset. The idea that the discrimination affinity is linear along a direction in space is restrictive. 5. Can you provide more details on the flipping experiment? Were NADs randomly permuted or only two of them flipped? I could not find the details in supplementary. 6. Let's take a network trained on CIFAR10. We find a test point with small (but non-zero) components along two top NADs. Now we flip their components or we kill one NAD component. Does it change the prediction of the network? How much does the image change visually? This experiment should capture the depencedence of a regularly trained CNN on NAD.

Correctness: Yes, but see above.

Clarity: Yes

Relation to Prior Work: Yes

Reproducibility: Yes

Additional Feedback: It's possible that I may not have understood all the details of CIFAR10 experiment (see point 4-6 in weakness section). I'll be happy to increase my rating if points to 4-6 are addressed in the rebuttal.


Review 4

Summary and Contributions: This work belongs to an important yet still underrepresented class of works aimed at understanding the structure of artificial neural networks rather than devising better ways to solve the downstream problems. The authors show that neural networks, in particular CNNs, are inherently anisotropic in that they have a much easier time finding discriminative features along some directions than along others. The authors show that this effect is due to the pooling layers that act as anisotropic lossy channels. The anisotropy affects not only the final discrimination quality but also the optimization process, slowing it down along "bad" directions. Therefore, it appears to be important to find an efficient way of learning these anisotropic directions (NADs). The authors provide an algorithm for this based on a simple test problem and Monte-Carlo sampling. The paper also contains a validation study that shows the anisotropic effect both for linearly separable datasets and for CIFAR-10 augmented with carrier features to test the effect of NADs. ===== I have read the authors' feedback and stand by my original (positive) assessment of the paper.

Strengths: The paper presents a completely novel (to the best of my knowledge) way to study neural networks and uncovers new structure in their behaviour. What's more, it not only recognizes the anisotropic effects but also provides constructive and efficient ways to find NADs. I believe that this is an important addition to our understanding of deep neural networks.

Weaknesses: One slight weakness is that I would prefer to see a more comprehensive evaluation study, in particular on larger and more practical datasets. If I understood the evaluation procedure correctly, it would not be prohibitively expensive to supplement CIFAR-10 with other datasets. Also, I would suggest that the authors expand upon several points that are interesting but remain unclear. First, the impact and applications of NADs. In particular, I am not sure I understood what the authors mean by incorporating prior knowledge about NADs into AutoML NAS loops: could you please elaborate on that? Second, it is clear from the paper that pooling layers are a source of anisotropy, but are there other sources? The authors give an example by removing all pooling layers from LeNet and ResNet18, but would a network that downsamples differently (e.g., the All Convolutional Net that has no pooling and downsamples by striding) be also free of this effect?

Correctness: As far as I could check, the proofs are correct (disclaimer: I did not check the proof of Lemma 1 in full). The experimental evaluation is sufficient to demonstrate the effect clearly.

Clarity: The paper is well-written and easy to follow. Minor: l.28: due <- due to l.153: that, trying ... random v, would be <- remove the commas l.275: understanding on <- understanding of

Relation to Prior Work: Yes, the prior art is adequately shown.

Reproducibility: Yes

Additional Feedback:

[Author Response · NeurIPS 2020]

We would like to thank the reviewers for their valuable feedback, which we will duly consider and integrate in our revised manuscript. According to reviewers, our paper presents "a completely novel way to study DNNs and uncovers new structure in their behaviour". Namely, we introduce the *new* notion of Neural Anisotropy Directions (NADs) "which are directions in the input space for which a CNN is biased to be able to easily separate". "What's more, we also provide constructive and efficient ways to find NADs" and validate their role on generalization on complex datasets. "This is an important addition to our understanding of DNNs" with "important insights for the NeurIPS/AI community".

**NADs and generalization** (R3) *The poison experiment, indeed, does not say much about the alignment of the NADs of CNNs with the features of CIFAR10, and the same behaviour is expected on other datasets.* The main insight in the poison experiment comes from the gradual increase in accuracy when the carrier is placed along higher NADs. Indeed, at the two extremes (first and last NADs) the DNN either picks the carrier or the CIFAR10 features. However, when the carrier is placed at the lower-middle NADs, the continuous increase in accuracy can only be explained if progressively more features from the CIFAR10 data are exploited by the DNN. We do not intend to imply that these features are linear, but that these (possibly non-linear) features can only depend on the NADs before the carrier, which highlights *the existence of an ordered preference of features for DNNs.*

Nevertheless, the experiment in Fig. 8 is precisely intended to demonstrate the alignment of NADs with the features of CIFAR10. In fact, flipping, i.e., reversing *all* the components of each image in the NAD basis, dramatically decreases the performance of these DNNs. Hence, since the flip is a lossless transformation, the only possible explanation is that the inductive bias of the DNN does not align properly with this new representation.

Despite the importance of analyzing the NADs for trained DNNs, the methodology proposed by R3 will unfortunately not work on this setup: The energy of CIFAR10 images is mostly concentrated on the first few NADs, and hence virtually no sample has small components in the first NADs. We are currently studying other avenues of research to test the dependence of trained DNNs on NADs.

**NADs beyond pooling** (R2,R4) Understanding the role of different architectural components in the shape of NADs is one of our principle active lines of research. In this regard, our preliminary experiments have identified that, beyond pooling and downsampling modules (e.g., striding), padding, skip connections and kernel sizes, can also heavily influence the qualitative behaviour of NADs. We plan to release a complete analysis of this problem in the near future.

**NADs and NAS** (R2,R4) We see Neural Architecture Search (NAS) as a longrun future application of our ideas, with two main flavours: The introduction of human perception priors, like a frequency or color preference profile; or the distillation of biases from one architecture to another, e.g., for network compression. The main challenge to use NADs on NAS is finding the way to introduce this prior in the optimization. A possible idea is to add a term in the objective that penalizes deviations from the target NADs.

**Other datasets for poison experiment** (R1,R3,R4) The main computational bottleneck for our validations on real data comes from the need to retrain a full DNN every time we poison the dataset. Unfortunately, the computational budget available to us does not allow to retrain a larger dataset that many times. We hope that the research community will be able to replicate our findings in these benchmarks as our code will be open sourced (R1).

**Poison perceptibility** (R3) The poison experiment is not designed to be imperceptible, but to test the preference of a DNN for different features. Our rough inspection suggests that perceptibility is influenced by image content, channel, and NAD index. In general, poisoning the first NADs seems more perceptible, but *does not change the image semantics*.

**Connection between Sec. 2 and 3** (R1) In Sec. 2, we provide a theoretical analysis of a simple model to illustrate how a particular layer can cause neural anisotropies. This is just an explanatory example, and does not provide a constructive way to identify NADs on a full architecture with multiple layers. Sec. 3 provides a tractable algorithm to compute NADs without explicitly requiring to know the mechanisms that generate them. Due to its complexity, we decided to leave the study of the complex interaction between layers for future work. We thank the reviewer for the suggestion.

**NADs and optimization** (R1,R2) NADs are not linked to a network instance or a specific training stage. Thus, they do not change during training. Instead, *they shape the optimization landscape and hence influence the training dynamics of an architecture*: When fitting a boundary on "good NADs" DNNs can do this faster than when fitting it on "bad NADs" and this can be seen in the speed of convergence of a DNN on different dirs. (see Fig.2 and Sec. D.3.1. in appendix).

**NADs computation** (R1,R3) Our method to identify NADs boils down to drawing multiple randomly initialized networks and performing a spectral decomposition on the resulting gradients. Because NADs specify a basis of the input space they have the same dimensionality as the input (e.g., 1024 for $32 \times 32$ pixels). The NAD index refers to the index of eigenvectors (R1). Overall, complexity is not an issue (R3), our method is in general very fast and scales to very large input dimensions. Furthermore, the main properties of interest are found on the first few NADs (recall the fast decay of eigenvalues), which are faster to compute. Finally, the number of classes at the final layer is not important: NADs are computed by replacing the final layer with a single output at which we compute the input gradient.

[Meta-Review · NeurIPS 2020]

The authors claim that NN architectures have directional inductive bias, which impacts test accuracy and training speed. They show how to compute such directions using spectral decomposition of network derivatives as a proxy for training speed. Consensus among the reviewers that this is an interesting direction to analyze inductive biases that are implicit in the network architecture.